bioengineering/biomedical engineering/
biomechanics

locomotion, stability, metabolic cost, Movement Amplification, centre of mass control

**Author for correspondence:**
Mengnan/Mary Wu
e-mail: mwissuper@gmail.com

# A novel Movement Amplification environment reveals effects of controlling lateral centre of mass motion on gait stability and metabolic cost

Mengnan/Mary Wu[1], Geoffrey L. Brown[1],
Jane L. Woodward[2], Sjoerd M. Bruijn[3]
and Keith E. Gordon[1,4]

[1]Department of Physical Therapy and Human Movement Sciences, Feinberg School of Medicine, Northwestern University, 645 N Michigan Ave, Suite 1100, Chicago, IL, USA
[2]Shirley Ryan AbilityLab, 355 E Erie, Chicago, IL, USA
[3]Department of Human Movement Sciences, Faculty of Behavioural and Movement Sciences, Vrije Universiteit Amsterdam, Institute for Brain and Behaviour Amsterdam and Amsterdam Movement Sciences, Amsterdam, The Netherlands
[4]Research Service, Edward Hines Jr. Veterans Administration Hospital, Hines, IL, USA

M/MW, 0000-0002-0341-4909

During human walking, the centre of mass (COM) laterally oscillates, regularly transitioning its position above the two alternating support limbs. To maintain upright forward-directed walking, lateral COM excursion should remain within the base of support, on average. As necessary, humans can modify COM motion through various methods, including foot placement. How the nervous system controls these oscillations and the costs associated with control are not fully understood. To examine how lateral COM motions are controlled, healthy participants walked in a 'Movement Amplification' force field that increased lateral COM momentum in a manner dependent on the participant's own motion (forces were applied to the pelvis proportional to and in the same direction as lateral COM velocity). We hypothesized that metabolic cost to control lateral COM motion would increase with the gain of the field. In the Movement Amplification field, participants were significantly less stable than during baseline walking. Stability significantly decreased as the field gain increased. Participants also modified gait patterns, including increasing step width, which increased the metabolic cost of transport as the field gain increased. These results support previous

research suggesting that humans modulate foot placement to control lateral COM motion, incurring a metabolic cost.

## 1. Introduction

During normal human forward walking, the body's centre of mass (COM) smoothly oscillates from side to side. Peak lateral excursions occur each step as the body weight is shifted toward a position above the supporting limb [1]. This lateral motion is a consequence of humans' preference to walk with non-zero step widths [2]; increasing step width results in proportional increases in lateral COM excursion [3]. To avoid falls during walking, COM position needs to be maintained, on average, within the base of support, defined by the centre of pressure of the feet on the ground [4–7]. If not controlled, this laterally directed COM motion may pose a threat to upright walking.

People use a combination of strategies to control lateral COM motion during walking [5]. One of the primary strategies is modulation of lateral foot placement [8,9]. On a step-by-step basis, variations in lateral foot placement are strongly associated with the COM state (position, velocity and acceleration) during the preceding stance phase [10,11]; for example, when the COM is falling faster towards the right, people take wider steps to the right. Taking a wider step will increase the safety factor distance between the COM position and lateral base of support [12] and also increase the frontal-plane gravitational moment acting about the ankle joint to arrest and then redirect the COM lateral velocity during single-limb support [8]. Thus, increasing step width may be beneficial for controlling lateral COM motion and increasing gait stability—the ability to reject or recover from small external perturbations and return to a steady state [13–17]. However, taking wider steps will also incur a metabolic energy cost related to increases in mechanical work required to redirect COM motion [2]. Research suggests that people select step widths that consider both metabolic energy cost and stability. During normal walking people select step widths that minimize metabolic energy costs [2], but in the presence of external perturbations that challenge stability, people often select wider and more energetically costly steps [18–23].

The strongest evidence for how people control lateral COM oscillations during walking and the associated metabolic energy costs come from observations of walking during conditions when the requirements to control lateral COM motion are reduced by applying external lateral stabilizing forces (i.e. stiff springs) to the pelvis [24,25]. In response to external lateral stabilization, people reduced step width, step width variability and metabolic cost of transport (COT) [24]. These findings suggest that during normal walking, control of lateral COM is partially regulated by lateral foot placement, and that this control has a metabolic consequence [24]. However, follow-up experiments using external lateral stabilization have not consistently yielded reductions in metabolic COT during normal walking conditions (i.e. with step width unconstrained) [26]. One criticism of this method is that the observed gait patterns may not have been purely a result of reducing the requirements to control lateral COM motion but rather emerged due to complex interactions with the novel stabilizing environment that also constrained potentially desirable pelvis motions [27]. While these past studies were both innovative and insightful, evaluating how people control lateral COM motion by reducing COM motion is an incomplete method to characterize control. To more fully characterize the strategies used to control lateral COM motion, complementary experiments should be performed that amplify lateral COM motion during gait. Demonstrating a continuum in the strategies used to control lateral COM motion and the associated metabolic costs across conditions of both reduced and amplified lateral COM motion during walking would strengthen previous conclusions.

Here, we introduce a novel experimental paradigm that was designed to increase the requirements for controlling lateral COM motion during walking. We use a cable-driven robot, the Agility Trainer, to apply continuous laterally directed forces to the pelvis during walking that are dependent on the user's own movements [28]. A Movement Amplification [29] environment, specifically a negative damping field, was created by modulating the external forces such that they were proportional in magnitude and in the same direction as the user's real-time lateral velocity. We hypothesize that compared to normal walking, people will take wider steps to aid in control of lateral COM motion in the Movement Amplification field, and that the strategies to control lateral motion will increase the metabolic COT.

## 2. Methods

### 2.1. Participants

Eighteen healthy individuals gave written informed consent prior to beginning the study. The Northwestern University and the Edward Hines Jr. Veterans Administration Hospital Institutional Review Boards approved the protocol. Inclusion criteria included: between 18 and 65 years of age, normal/corrected vision, able to walk continuously for 10 min without undue fatigue or health risks, and free from musculoskeletal, neurological or balance impairments limiting ambulation. Data from three participants were excluded due to data collection errors (two participants) and psychological discomfort performing the required treadmill walking (one participant). The 15 participants analysed were $24 \pm 4$ years, height $1.73 \pm 0.06$ m, body mass $84.8 \pm 30.2$ kg and 5 males/10 females.

### 2.2. Experiment protocol and set-up

To measure pelvis and foot kinematics, we tracked a total of 19 reflective markers placed on the pelvis (anterior superior iliac spines, iliac crests and three tracking markers) and bilaterally on the greater trochanter, lateral knee, lateral malleolus, calcaneus, and second and fifth metatarsals. We recorded three-dimensional marker coordinates at 100 Hz using an 11-camera motion capture system (Qualisys, Gothenburg, Sweden). We also collected synchronized kinetic data from the Agility Trainer's load cells at 1000 Hz.

Participants walked on an extra-wide treadmill—belt width 1.39 m (Tuff Tread, Willis, TX)—providing space to safely perform lateral corrective steps [30]. Participants wore a trunk harness attached to a passive overhead safety support system (Aretech, Ashburn, VA) that provided no bodyweight support and was adjusted to allow unrestricted travel across the treadmill. Each participant's preferred walking speed was identified through a staircase method of increasing and decreasing the treadmill speed until desired speed was confirmed through verbal feedback. Participants then walked for 2 min to acclimate to their preferred treadmill speed. The preferred treadmill walking speed was $1.18 \pm 0.09$ m s$^{-1}$ across all participants. All further walking for the experiment was done at this preferred speed.

Participants then donned the portable metabolic unit (COSMED, Chicago, IL, USA) and stood with minimal motion and no external support for 6 min while we recorded oxygen consumption. To measure metabolic energy expenditure during the experiment, we recorded breath-to-breath oxygen consumption using a portable gas analysis system (COSMED).

Next, to create a consistent challenge level for the walking task across participants, we identified participants' baseline peak-to-peak lateral COM excursion per stride and used this information to determine the width of the participant-specific visual target lane that would be used during experimental trials. During this procedure, participants walked while watching the visual feedback marker (a black dot) of their real-time lateral COM position oscillate laterally on the screen. They were instructed to walk normally, allowing the marker to oscillate but centred about a vertical black line in the middle of the screen. An algorithm based on pilot studies set an initial invisible lane width of 6.9 cm centred about the vertical line and increased lane width by 3.4 cm when the COM position crossed either border of the lane 20 times. After an increase in lane width, the number of lane border crossings reset to zero. After participants maintained their COM position within the lane for 20 consecutive seconds, the resulting invisible lane width was multiplied by 1.5 and used as the target lane (figure 1a) width for the main experiment. Participants were not given any information about the calculation of the target lane width. Across all participants, the target lane width was $0.13 \pm 0.03$ m.

Participants then completed four 12 min walking trials: *Null, Amplification Low, Amplification High* and *Stabilization*. During amplification and stabilization trials, a custom-built cable-driven robot, the Agility Trainer, applied continuous lateral forces to the pelvis during treadmill walking [28]. Independent series-elastic linear motors (Baldor, Fort Smith, AR) created bidirectional forces that were transmitted via steel cables to medial attachments of a snug pelvic harness (Barry, Montreal, Quebec). A pair of load cells (Omega, Norwalk, CT) connected in series with the cables measured the applied forces. We have previously described in detail how people respond to a stabilizing viscous force field [30]. As such, we do not analyse or discuss the response to the Stabilization field within the current experiment. The order of the trials was randomized. For each trial, participants completed two 5 min walking bouts in a given force field separated by a 2 min standing rest period (figure 1b). Pilot data suggested that people would reach steady state in the novel amplification force fields by the end of 5 min of walking.

(*a*) real-time visual feedback

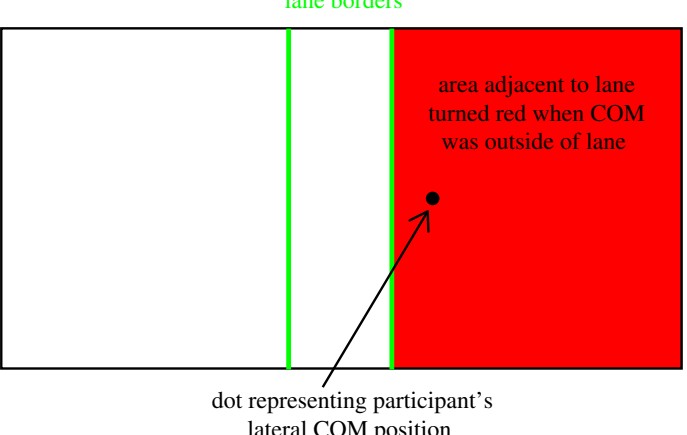

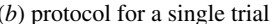

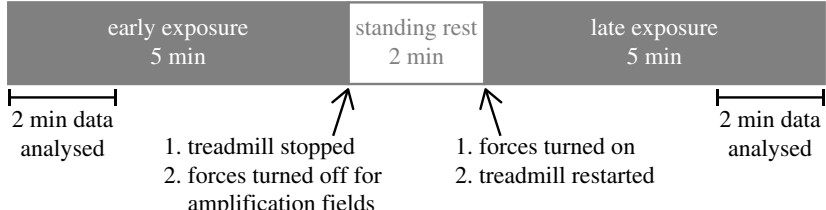

**Figure 1.** (*a*) Visual feedback for the walking task displayed on a screen in front of the treadmill. Participants were instructed to control their walking such that the black dot representing their real-time lateral COM position was maintained within the borders of the target lane. When COM position was within the lane borders, the lane turned green; when COM position was outside the borders, the area adjacent to the lane turned red. The width of the lane for each participant was determined by their baseline mediolateral COM excursion during preferred-treadmill-speed walking. (*b*) Experiment protocol for one trial. The order of fields (Null, Amplification Low, and Amplification High) was randomized. Data from the first 2 min or 200 steps of the early exposure period were analysed to verify that positive net work was done in the amplification fields. Data from the last 2 min or 200 steps of the late exposure period were analysed to compare stability, metabolic cost and gait patterns across force fields.

We doubled the total walking for this experiment to ensure that participants had sufficient time to fully adapt to the force fields. At the completion of the second walking bout of each trial (with the exception of the Null trial), the applied forces were removed and participants walked for an additional 2 min to wash out any residual effects of the force field, based on previous experiments showing after-effects in a damping field [30]. Participants then took a 2 min standing rest before beginning the next walking trial.

During the *Amplification Low* and *Amplification High* fields, participants experienced a lateral force proportional in magnitude and in the same direction as their real-time lateral COM velocity. These negative damping force fields amplified lateral motion with the objective to increase the difficulty of maintaining straight-ahead walking. Damping gains were $-45$ and $-55\,\mathrm{N\,s\,m^{-1}}$ for the *Amplification Low* and *Amplification High* fields, respectively. For safety, we capped the forces at 80 N; however, this force was never reached during the experiment. The *Amplification High* gain was selected as the largest negative damping gain that could be reliably created with the motor control system [28]. We selected the value of the magnitude of the negative damping gain during the *Amplification Low* field based on pilot studies where we ramped up the value until participants felt a noticeable effect of amplification. During the *Null* field, no forces were applied to the participant.

There is a complex relationship between a participant's anthropometrics (e.g. mass, inertia, leg length) and continuous lateral COM velocity that influences the resulting lateral acceleration of their COM created by each force field. As this was our first attempt to quantify the effects of the movement amplification force fields on COM dynamics, we chose not to scale the force field gains for each participant. While there will probably be absolute differences in the resulting COM dynamics experienced between individuals with different body mass [31], we assumed that the two gains chosen were sufficient to test changes in control strategies and associated metabolic energy costs when walking in a low- and high-gain negative damping field.

During all trials, we instructed participants to control their walking such that a black dot representing their real-time lateral COM position on a visual feedback display was maintained within the borders of

the target lane (figure 1*a*). This task was designed to motivate participants to expend effort to control their COM, which helped us to distinguish whether changes in lateral COM motion were due to an *inability* to control COM motion as opposed to a deliberate choice to not control COM motion.

To provide this visual feedback, a 60-inch screen mounted 1.8 m in front of the treadmill provided continuous real-time visual feedback of the participant's lateral COM position during walking. The motion capture system streamed three-dimensional marker locations of the two greater trochanter markers in real time to a custom LabVIEW virtual instrument (National Instruments, Austin, TX) that created the visual display. The virtual instrument estimated lateral COM position as the midpoint of the two greater trochanter markers. As the participant moved left or right, a visual feedback marker (a black dot), moved simultaneously on the screen using approximately 1 : 1 scaling (figure 1*a*). To limit cognitive demand on the participant, only lateral position information was displayed. Depending on the protocol section, additional visual information—either a vertical line or target lane—was overlaid with the visual feedback marker.

As we were primarily interested in how people modulate stepping characteristics to control lateral COM motions, we also instructed participants to walk with their arms crossed to minimize the contributions of the upper extremities. Not allowing arm swing has been shown to increase metabolic cost of walking [32–34], so we chose to include this restriction to make the effect consistent across participants as opposed to allowing participants to choose whether or not to use arm swing. Participants were not provided with handrails or any other external assistive devices and were instructed to walk as they felt most comfortable within these guidelines.

We collected metabolic data continuously for the duration of each trial. We collected kinematic and kinetic data during early and late exposure periods, defined as the first and last 2 min of the walking trial, respectively (figure 1*b*).

## 2.3. Data processing and calculations

To estimate the energetic requirements of walking in the different force fields, we calculated metabolic COT during the late exposure period, when participants had reached steady state. Average metabolic power (W) during each trial was calculated using the standard relationship of 20.9 W for each millilitre of oxygen consumed per second [35]. To calculate the metabolic power of walking, the average metabolic power during quiet standing was subtracted from the average metabolic power during walking. COT was calculated by normalizing the metabolic power of walking to each participant's body mass and preferred walking speed.

We examined kinematic marker data to quantify gait adaptations in response to the force fields and kinetic load cell data to calculate net work performed on the participant. Kinematic and kinetic data were analysed during the first 200 steps of early exposure and during the last 200 steps of late exposure (figure 1*b*). For all participants, the 200 steps analysed during the late exposure period occurred within the 2 min period used to analyse COT. We fixed the number of steps used to analyse kinematic and kinetic data as opposed to time length since calculation of our primary measure of stability, the short-term local divergence exponent, is sensitive to the number of steps [36].

Kinematic data were processed using Visual3D (C-Motion, Germantown, MD) and a custom Matlab (Mathworks, Natick, MA) program (all Matlab code used for data processing, as well as the dataset are located at https://osf.io/5j7rk/). Marker data were gap-filled and low-pass filtered (Butterworth, 6 Hz cut-off frequency). Load cell data were filtered with the same low-pass filter as marker data. Time of initial foot contact (IC) and toe-off (TO) events were identified for each step based on local extrema of fore-aft positions of the calcaneus and second metatarsal markers [30,37]. Mediolateral COM position was calculated in Visual3D as the centre of the 'Visual3D pelvis' model [38]. Although this method will not account for relative changes in COM position within the body, this estimate should give a reasonable approximation of true COM motion, since arm motion was restricted in this study and a single marker placed at the second sacral vertebra has been shown to highly correlate with lateral COM of the body during walking, slips and turns [39,40]. COM velocity was calculated as the derivative of COM position.

First, to verify that the amplification fields were amplifying participants' COM motion as intended, we calculated the average net work performed by the field on each participant per stride by integrating instantaneous power (COM velocity multiplied by net force) in the lateral direction and normalizing by the participant's body mass and number of strides (100) taken. Positive net work values indicate that the field's applied forces were in the same direction as the participant's COM lateral velocity, which is consistent with the design of the field to amplify movement. One participant's data were excluded from analysis of net work due to a data acquisition error.

For our metric of stability, we calculated local stability of lateral COM velocity. We chose lateral COM velocity since walking is passively unstable in the mediolateral direction (1–3) and velocity is not affected by non-stationarities, as opposed to position data. The maximum Lyapunov exponent ($\lambda$) quantifies the average logarithmic rate of divergence of a system after a small perturbation [41,42] and can be calculated for short and long time periods between states. The short-term local divergence exponent ($\lambda_s$) has demonstrated theoretical and predictive validity in simulation and empirical studies of walking [13]. Due to sensitivity to the number of steps, this metric was calculated from a fixed number (200) of steps [36]. For construction of a state space, we used a time delay of 10 samples and an embedding dimension of 5. Within this state space, nearest neighbours were identified, and their divergence tracked. From these distances, a log(divergence) curve was calculated, and the local divergence exponent was calculated as the slope of this curve between 0 and 0.5 strides [43].

We calculated means and variability of several kinematic metrics to investigate the gait strategies employed in the different force fields. We examined mean minimum lateral margin of stability (MOS) as well as the mean and variability of: peak lateral COM speed, step width and step length using methods described previously [44] with the exceptions of using the lateral malleolus markers for step width calculations. In addition, we calculated double-support time (DST) as the mean percentage of a stride spent in double-limb support. One participant's data were excluded from the MOS and DST metrics due to poor marker coverage during some trials.

Finally, we regressed lateral foot position to lateral COM (estimated from the pelvis) velocity and position at the previous step's midstance based on an existing model [10]. Regression coefficients and quality of fit ($R^2$) were calculated for each participant in each force field for the early and late exposure periods separately. The $R^2$-value of the regression fit was a metric of the degree of coupling between foot placement and COM dynamics, with higher correlation suggesting either increased neural control or effects of passive dynamics [10].

## 2.4. Statistical analysis

Net work done by the field on the participant was analysed during early exposure to verify that the amplification fields performed as intended before adaptation, and during late exposure to examine participants' final adapted behaviour in the field. To verify that the fields were amplifying participants' COM motion, we performed separate two-tailed $t$-tests (or Wilcoxon sign-rank tests if normality was violated) to compare if net work in the amplification fields were significantly different from a value of zero. Another two-tailed $t$-test was performed to compare if net work was significantly different between the Amplification Low versus High fields to verify that the amplification magnitudes were significantly different. Net work was zero for the Null field since the Agility Trainer was turned off and the load cells attached in series hung slack. Bonferroni corrections were used for these three pairwise comparisons, and significance was set at the $p < 0.05$ level (i.e. the unadjusted pairwise comparison $p$-values were multiplied by 3 and then compared versus 0.05). Data from one participant were excluded from the early exposure data due to poor force signal quality upon visual inspection.

All other statistical tests focused on participants' final adapted behaviour during late exposure. To compare differences between fields, we performed separate one-way repeated measures ANOVAs for metrics of stability ($\lambda_s$), metabolic COT and gait kinematics (mean peak lateral COM Speed, peak lateral COM speed variability, mean step width, step width variability, regression $R^2$, step length, step length variability, MOS and DST). If sphericity was violated, the Greenhouse–Geisser $F$-statistic and $p$-value were used to test the main effect of the field. If assumptions of the parametric repeated measures ANOVA model were violated, Friedman's test was used instead. When a significant main effect was found from either the repeated measures ANOVA or Friedman's test, Bonferroni-corrected pairwise comparisons were made between all fields (i.e. the unadjusted pairwise comparison $p$-values were multiplied by 3 and then compared versus 0.05). Significance was set at the $p < 0.05$ level for all tests and comparisons.

To further investigate metabolic results, non-significant pairwise differences between fields for COT were examined with equivalence tests for differences between dependent means with a medium effect size of $dz = 0.35$ and $\alpha$-value of 0.05 [45]. While not often used in biomechanical research, equivalence testing is becoming more common in other fields. It allows one to make statements about the probability of finding the experimental data when assuming that in reality there is a certain minimal effect practical for investigation [45]. As such, it can be seen as a test for the original null hypothesis.

# 3. Results

## 3.1. Verification of Movement Amplification fields

During early exposure, net work performed by the amplification field on the participant was significantly ($p < 0.05$) positive in both Amplification Low ($0.0128 \pm 0.0126$ J/(stride × kg)) and Amplification High ($0.0397 \pm 0.0192$ J/(stride × kg)) fields. These results, along with visual observation (video S1 at https://mfr.osf.io/render?url=https://osf.io/5j7rk/?action=download% 26mode=render and figure 2), confirmed that the amplification fields operated as intended during early exposure, pulling the person's body in the direction of their movement. Net work was also significantly ($p < 0.05$) larger (210%) in the high versus low field, showing a difference between amplification magnitudes before participants had an opportunity to adapt gait patterns.

During late exposure, net work was significantly ($p < 0.05$) positive for the Amplification Low ($0.0127 \pm 0.0104$ J/(stride × kg)) and Amplification High ($0.0244 \pm 0.0167$ J/(stride × kg)) fields, confirming that forces were applied in the direction of COM velocity. Net work was also significantly ($p < 0.05$) larger (92%) in the high versus low field, showing a difference between amplification magnitudes during late exposure (figure 3).

## 3.2. Stability and metabolic cost

During late exposure, participants were significantly ($p < 0.05$) less stable in both amplification fields (13% and 20% in the Amplification Low and Amplification High fields, respectively) versus Null and also significantly ($p < 0.05$) less stable in Amplification High versus Amplification Low (6.1% higher (less stable)). COT was significantly ($p < 0.05$) higher (by 18%) in the Amplification High field versus Null (figure 4b). Since COT was significantly ($p < 0.05$) higher (by 9.7%) in the Amplification High versus Low field, the non-significance between Amplification Low and Null suggested that either baseline metabolic cost was maintained during Amplification Low or that the comparison lacked statistical power. A follow-up equivalence test revealed that the pairwise comparison was probably underpowered ($t = 0.535$, $p = 0.70$), so equivalence could not be confirmed.

## 3.3. Kinematics

During late exposure, participants exhibited significant differences in mean gait kinematics in the amplification fields versus Null (figure 5). In both amplification fields, participants showed slower peak lateral COM speed, wider steps and larger MOS. Mean peak lateral COM speed was 16% and 14% slower ($p < 0.05$) in the Amplification Low and Amplification High fields, respectively, versus Null (figure 5a). Step width was 5.1% and 8.2% wider ($p < 0.05$) in the Amplification Low and Amplification High fields, respectively, versus Null (figure 5b). Similar in trend to step width, MOS was 44% and 53% larger ($p < 0.05$) in the Amplification Low and Amplification High fields, respectively, versus Null (figure 5c). Step length was 3.5% shorter ($p < 0.05$) in the Amplification High field versus Null (figure 5d). Finally, DST was 3.5% shorter ($p < 0.05$) in the Amplification High field versus Null (figure 5e).

Kinematic variability and coupling between foot placement and COM dynamics significantly increased in the amplification fields versus Null (figure 6). Peak lateral COM speed variability was 73% higher ($p < 0.05$) in Amplification High versus Null (figure 6a). Step width variability was 51% and 114% higher ($p < 0.05$) in the Amplification Low and Amplification High fields, respectively, versus Null (figure 6b). Lateral foot placement was significantly more correlated to COM dynamics ($p < 0.05$); regression $R^2$ was 0.80 and 0.82 in the Amplification Low and High fields, respectively, versus 0.71 in Null (figure 6c). Step length variability was 108% higher ($p < 0.05$) in the Amplification High field versus Null (figure 6d).

Kinematic results also differed significantly between the two magnitudes of the movement amplification field. Participants took 2.3% shorter steps (figure 5d) and spent 1.9% less time in double-limb support (figure 5e) in Amplification High versus Low ($p < 0.05$). Step width (figure 6b) and step length (figure 6d) were also significantly ($p < 0.05$) more variable, 42% and 44%, respectively, in the high versus low field.

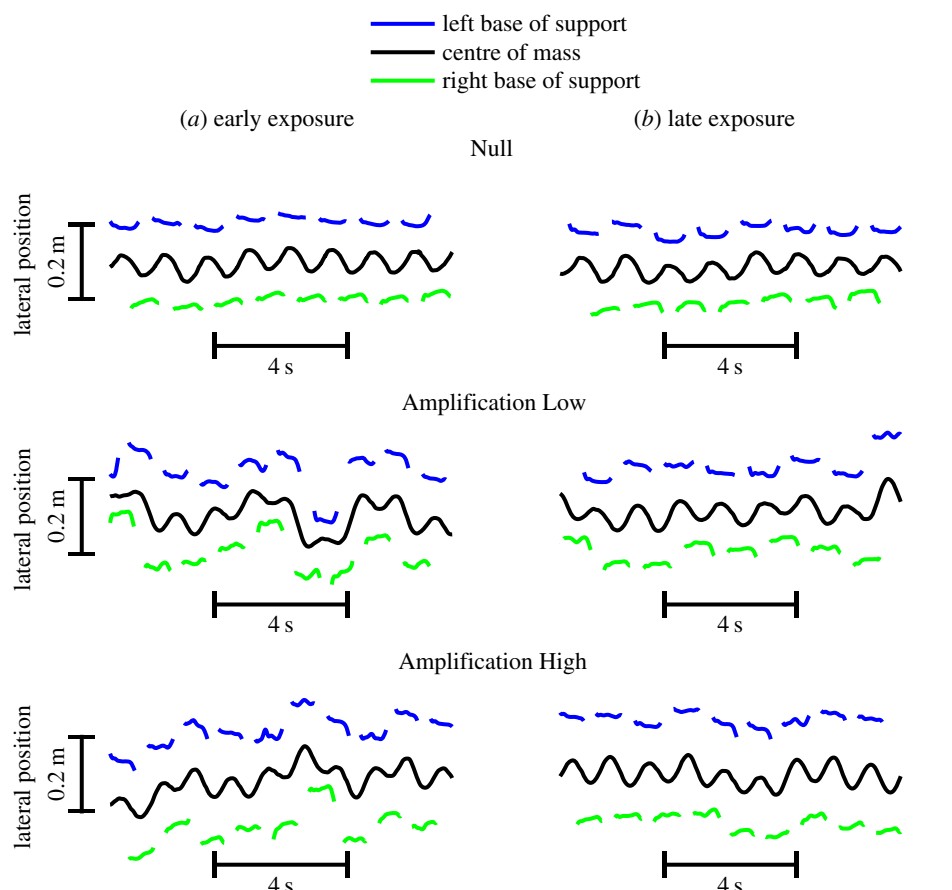

**Figure 2.** Lateral centre of mass position and base of support (lateral malleolus marker) versus time for sample participant during (*a*) early and (*b*) late exposure for all force fields (Null, Amplification Low and Amplification High).

# 4. Discussion

## 4.1. Changes in metabolic cost and stability scale with the gain of the Movement Amplification field

The novel Movement Amplification field introduced in this study allowed us to examine how people control lateral COM motion during walking. As the gain of the field and the corresponding challenge to control lateral COM increased, we hypothesized that people would respond by modulating foot placement to regulate COM motions, and that these control strategies would incur a metabolic cost. Outcomes from this study supported our hypotheses. Increasing the gain of the Movement Amplification field resulted in significant and progressive reductions in local stability. Participants changed their foot placement patterns in response to the Movement Amplification field gains. As the gain increased, so did step width, step width variability and step length variability. Step length decreased in the Movement Amplification field. In addition, the correlation between lateral foot placement and COM state were stronger in the Movement Amplification field than during normal walking. Collectively, these changes in foot placement in the Movement Amplification field probably assisted in controlling lateral COM motion by increasing the lateral MOS and decreasing peak COM velocities when compared to normal walking. The strategies used to control lateral COM motion tended to increase the metabolic COT, with the greatest metabolic costs occurring in the high-gain Movement Amplification field. Thus, we found that people modify gait patterns in response to increasing challenges to control lateral COM motions, and that these control strategies require metabolic energy.

Interestingly, metabolic cost was not significantly different between the Amplification Low field and Null. Equivalence testing revealed that this comparison was underpowered; therefore, it is inconclusive whether the Amplification Low field resulted in changes in energetic costs versus baseline walking. The trend we observed when metabolic cost was plotted relative to the amplification field gain suggested a

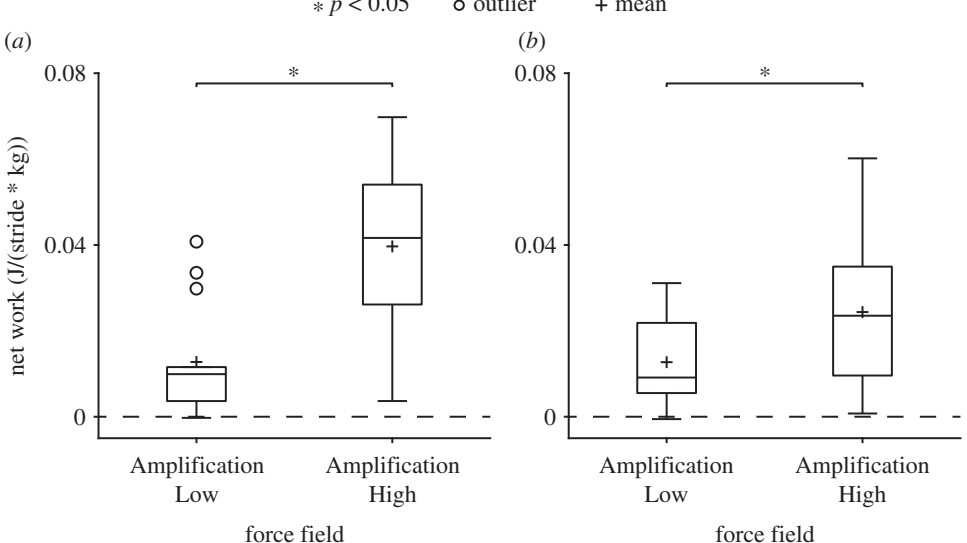

**Figure 3.** Net work performed on the participant's COM by the force field during (*a*) early and (*b*) late exposure for all participants in the Amplification Low and Amplification High fields. Net work was zero for the Null field. Net work in Amplification Low and High fields were significantly greater than zero and significantly different from each other for both early and late exposure.

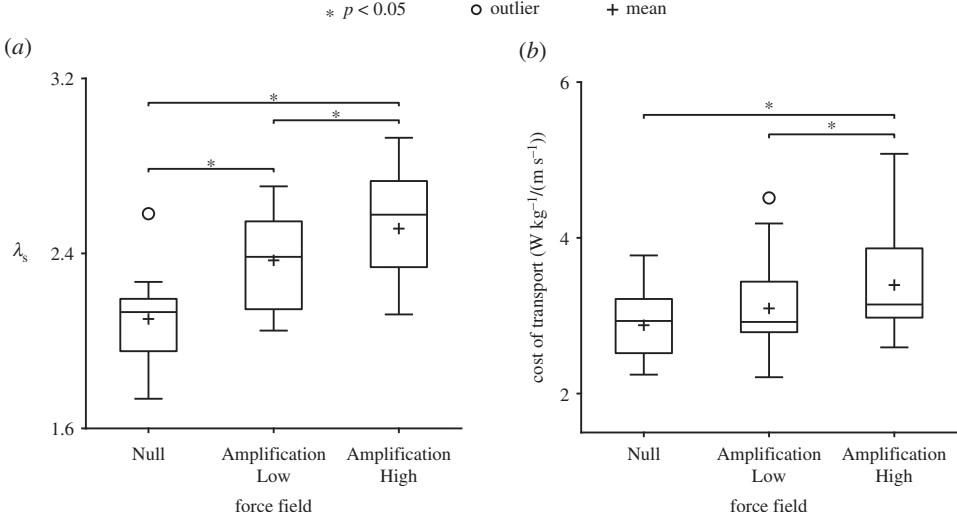

**Figure 4.** Stability and metabolic COT for all participants during late exposure for each force field. (*a*) Participants' COM velocities were significantly less stable, i.e. resulted in a larger short-term local divergence exponent ($\lambda_s$), in both amplification fields versus Null and also significantly less stable in Amplification High versus Amplification Low. (*b*) COT was significantly higher in Amplification High versus Null and in Amplification High versus Low.

linear trend. However, if the two conditions were in fact equivalent, these results could potentially suggest that in certain low-threat situations, individuals may be willing to sacrifice some gait stability (which was significantly reduced in Amplification Low versus Null) in order to maintain a lower metabolic cost.

## 4.2. Changes in foot placement and other gait patterns suggest a combination of control strategies were employed in the Movement Amplification fields

Our findings are consistent with results from previous studies examining control of lateral COM motion during gait. Previous research has found that when the requirements for controlling lateral COM motion during gait were *decreased*, people adopted narrower and less variable step widths [24–26,46]. In the current study, in which the challenge to control lateral COM motion during gait was *increased*, foot placement changed in the opposite direction of the earlier studies. We found that in response to the

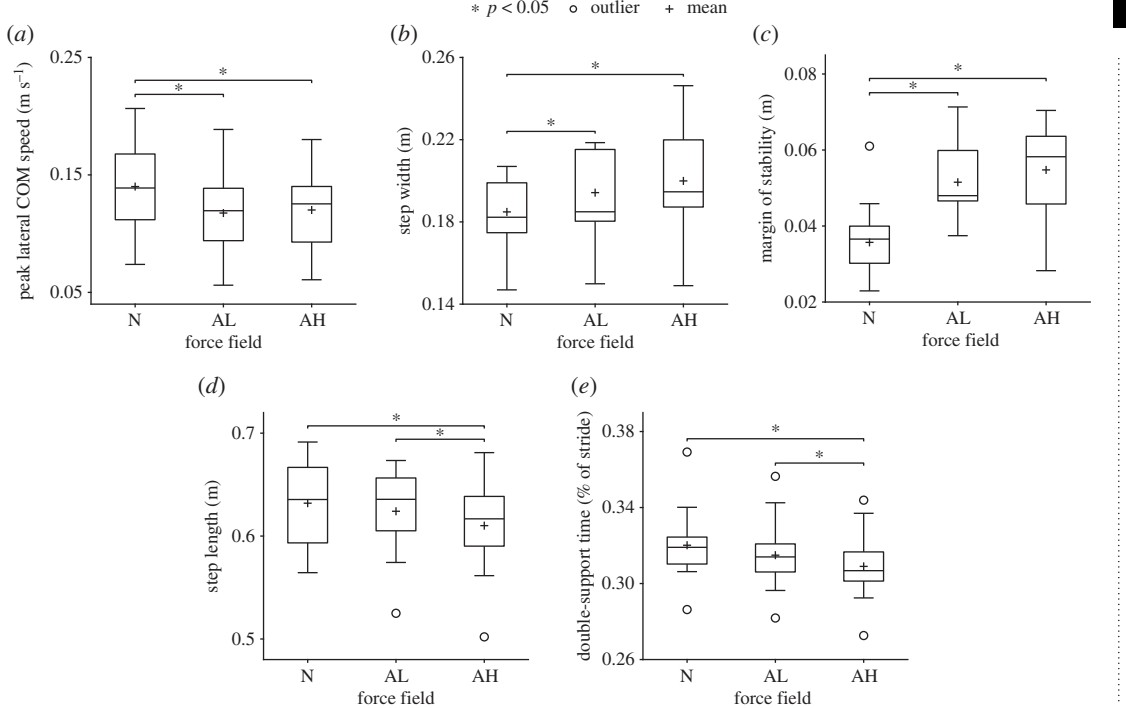

**Figure 5.** Mean kinematics for all participants during late exposure for each force field ('N' = Null, 'AL' = Amplification Low, 'AH' = Amplification High). (*a*) Mean peak lateral COM speed was significantly lower in both amplification fields versus Null. Both step width (*b*) and minimum lateral MOS (*c*) were significantly higher in both amplification fields versus Null. (*d*) Step length was significantly shorter in the Amplification High field compared to Null and significantly shorter in Amplification High versus Low. (*e*) Double-support time was significantly lower in Amplification High versus Null and significantly lower in Amplification High versus Low.

increased challenge to control COM motion, step width was wider and more variable than normal walking. Wider steps may have increased control of lateral COM motion by increasing the base of support, which in turn may have increased the lateral MOS, a safety factor that considers the distance and velocity of the COM relative to the base of support [12].

The increases in step width variability may have also contributed to the instantaneous control of lateral COM motion. In the Movement Amplification fields, we observed increased variability of lateral COM speed and a stronger relationship (larger $R^2$) between COM state and lateral foot placement location. The increases in step width variability may have reflected more step-by-step changes in lateral foot placement that were coordinated with instantaneous changes in COM state; e.g. wider steps may have been taken when lateral velocity was greater. However, as others have expressed, it is not possible to discern from the current experiment if the observed change in relationship between COM state and foot placement is a result of active neural control or a result of passive dynamics [4,10].

It is likely that these changes in step width were used in coordination with other control strategies. For example, we also observed that individuals selected shorter step lengths as the Movement Amplification field gains increased. Because treadmill speeds were constant, taking shorter steps resulted in high-cadence stepping patterns. This could be valuable for two reasons: first, shorter steps reduce the average peak velocities of the COM, and second, faster cadences allow corrective steps to be implemented more frequently over a fixed period of time [5]. Similarly, participants spent less time in double-limb support during Amplification High versus Low, which may have been valuable for increasing the probability at any instant that a limb was in swing phase and available to rapidly make a corrective step if necessary.

## 4.3. Differences in metabolic cost between Movement Amplification field gains may be explained by mechanical work

The difference in metabolic cost between the low and high Movement Amplification fields may be explained by differences in mechanical work performed *by the person* in the frontal plane. Since Movement Amplification only does positive work on the COM, to maintain a similar COM trajectory as baseline walking, the person needs to perform greater negative work to redirect COM velocity

∗ *p* < 0.05    ○ outlier    + mean

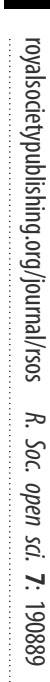

**Figure 6.** Kinematic variability and coupling between COM and foot placement for all participants during late exposure for each force field ('N', Null; 'AL', Amplification Low; 'AH', Amplification High). (*a*) Peak lateral COM speed was significantly more variable in Amplification High versus Null. (*b*) Step width was significantly more variable in both amplification fields compared to Null, and also significantly more variable in Amplification High versus Low. (*c*) Regression $R^2$-values were significantly greater in Amplification Low and High versus Null, showing that lateral foot placement was more closely correlated to COM dynamics in the amplification fields. (*d*) Step length was significantly more variable in Amplification High versus Null and significantly more variable in Amplification High versus Low.

during step transitions (figure 7). In this study, we observed increases in mean step width which allow for an increased ability to perform negative work at the COM [2]. People may also perform less positive work to recover energy lost to collision costs and rely instead on the Movement Amplification field to do positive work for them (i.e. the Movement Amplification field would increase the positive velocity in single-limb support in figure 7*b*). Given that negative work has higher metabolic efficiency than positive work [47], the Amplification Low field may have resulted in small net changes to metabolic cost by assisting with positive work and slightly increasing the amount of negative work performed by the person, even though step width increased compared to baseline walking. However, for the Amplification High field, the ratio between positive work savings and increased negative work demand probably changed, and the additional energy required for maintaining lateral stability with negative work might have resulted in a net metabolic cost exceeding that of baseline walking. Future work measuring ground reaction forces for each limb would allow us to verify these hypotheses.

## 4.4. Local stability and margin of stability describe different concepts of 'stability'

The seemingly counterintuitive results for $\lambda_s$ versus MOS metrics demonstrate the need to carefully interpret the MOS metric, which is primarily a measure of passive stability of the whole body

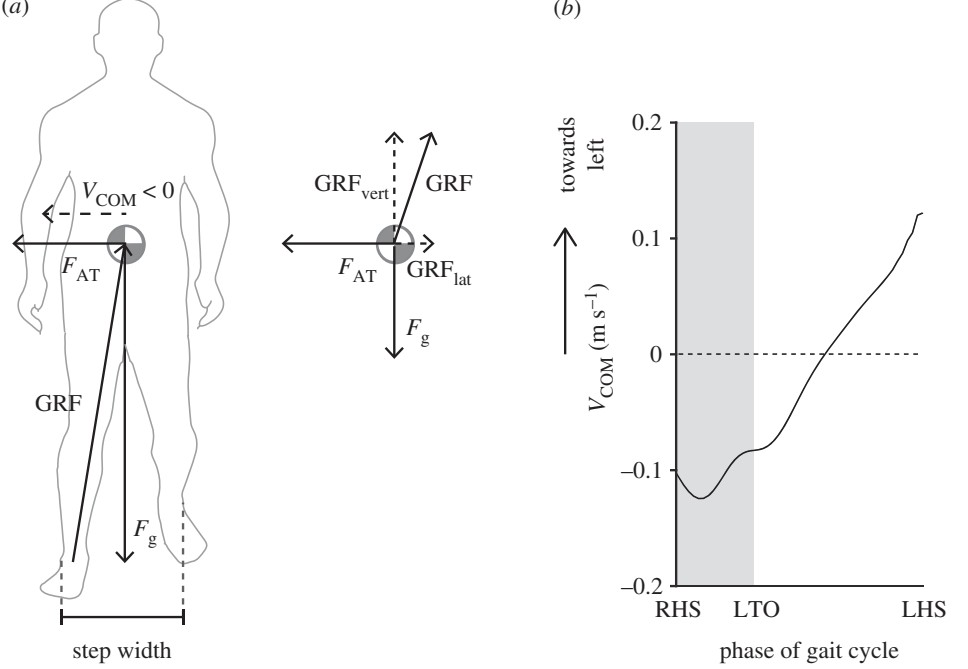

**Figure 7.** (a) Gravitational force ($F_g$), ground reaction force (GRF), lateral velocity of COM ($V_{COM}$) and force from the Agility Trainer ($F_{AT}$) during early right single-limb support when COM velocity is directed laterally (defined as > 0 towards the participant's left). (b) Velocity of COM versus phase of gait cycle for a sample participant showing reversal of direction of $V_{COM}$ during the middle of single-limb support (grey denotes double-support phase defined from right heel strike (RHS) to left toe-off (LTO)).

modelled as a pendulum [12] and has been shown, counterintuitively, to increase with destabilizing lateral perturbations [30,48]. Participants in the current study might have increased this passive pendulum stability about the COM by increasing MOS in the amplification fields, but they still had higher variability and lower local stability in COM velocity. This implies that people remain within their base of support to maintain forward walking via pendulum stability, but COM dynamics are less locally stable. As such, MOS might be more appropriately interpreted as a description of a strategy employed in response to balance challenges, in contrast to $\lambda_s$, which describes the convergence of a system in response to small perturbations. It is interesting to note that while the mean peak lateral COM speed decreased in the amplification fields, the variability and $\lambda_s$ of COM velocity increased. This result is consistent with previous work showing that slower forward-progression walking speed is not necessarily more locally stable [43,49].

As a limitation, $\lambda_s$ was constructed as a stability metric for 'small' perturbations during periodic motion [13], which we assumed that our protocol satisfied.

## 4.5. Movement Amplification for locomotor training

The novel Movement Amplification environment used in the current experiment to challenge control of lateral COM during walking may have clinical applications for retraining locomotor stability deficits in a different manner than applying unexpected or unpredictable perturbations to the body [21,23,30,50] or walking surface [19,22]. The Movement Amplification environment is similar to the paradigm of Error Augmentation, in which a person performs movements in an unstable robotic force field that magnifies self-generated motion [51]. Error Augmentation may accelerate motor learning by augmenting sensory-motor feedback [52] and allowing the impaired nervous system to identify otherwise imperceptible errors. The intensified feedback may aid an individual with sensory-motor impairments to recalibrate their internal model so that it can be used to control anticipatory components of movement. Importantly, there is evidence that Error Augmentation can be used successfully for accelerating the adaptation of desired walking patterns [53]. Movement Amplification is a more general environment than Error Augmentation and does not use a reference trajectory to identify errors, but instead uses negative impedance to amplify all movements (not just errors) [29]. Movement Amplification may also aid in the acquisition of new motor patterns by encouraging movement exploration [54].

While Error Augmentation and Movement Amplification approaches have shown promise for retraining control of reaching movements in impaired populations [29,51], these methods have not been applied to training locomotor stability. In the current experiment, within 10 min, participants had adapted their steady-state walking patterns in response to the challenge of controlling their lateral COM motion in the Movement Amplification force field. This suggests that Movement Amplification may be a feasible paradigm to incorporate into locomotor training interventions targeting gait stability.

## 5. Conclusion

When the Movement Amplification gain was high, participants were less locally stable and expended significantly more energy than baseline walking. When the Movement Amplification gain was low, participants were less locally stable and trended toward an increase in metabolic cost compared with baseline walking. Participants adapted different gait patterns in response to varying magnitudes of Movement Amplification, increasing step width and the step-by-step relationship between COM state and lateral foot placement. These results expand on outcomes from previous studies and suggest that control of lateral COM motion during walking is controlled in part by modulations in foot placement, and that as the challenges to control COM motion increase, there are associated increased metabolic costs.

Ethics. All participants gave written informed consent prior to beginning the study to either the Research Physical Therapist, Jane Woodward or the Principal Investigator, Dr. Keith Gordon. The Northwestern University Institutional Review Board approved the protocol: IRB# STU00071150. The Edward Hines Jr. Veterans Administration Hospital Institutional Review Board approved the protocol: IRB #15-011.
Data accessibility. The supplementary video, datasets supporting the conclusions of this article, and Matlab code for analysis are available in the Open Science Network repository, https://osf.io/krvgx/.
Authors' contributions. M.W., G.B. and K.G. designed the experiment protocol. M.W., J.W. and K.G. coordinated the study. M.W., G.B., J.W. and K.G. collected data. J.W. conducted clinical exams. G.B. and K.G. designed the agility trainer device. M.W. conducted data analysis and statistical analyses. S.B. assisted with data and statistical analysis. M.W. and K.G. drafted the manuscript. G.B., S.B. and J.W. were involved in critically editing the manuscript. All authors gave final approval for publication and agree to be held accountable for the work performed therein.
Competing interests. We declare we have no competing interests.
Funding. Supported by Career Development Award 2 #1 IK2 RX000717–01 and Merit Review Award # I01RX001979 from the United States Department of Veterans Affairs, Rehabilitation Research and Development Service.
Acknowledgements. The authors thank Chelsea Rugel and Grace Bellinger for assistance processing data and the Human Agility Lab members for their feedback and edits on this manuscript.

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
