## [Reviewer comments · Royal Society Open Science]

Review History

RSOS-190889.R0 (Original submission)

Review form: Reviewer 1 (Jason R. Franz)

Is the manuscript scientifically sound in its present form?

Yes

Are the interpretations and conclusions justified by the results?

No

Is the language acceptable?

Yes

Do you have any ethical concerns with this paper?

No

Have you any concerns about statistical analyses in this paper?

No

Recommendation?

Major revision is needed (please make suggestions in comments)

Comments to the Author(s)

This study uses a novel lateral force field to test for the presence of trade-offs between stability and metabolic energy cost during walking in the absence of costs one might expect due to anticipatory changes arising from a threat to balance. The experimental approach is innovative, and the outcomes are interesting and relevant to the field of biomechanics and motor control. My most significant concern (see below) is that the results tend not to convincingly support the conclusions. I outline below several opportunities the authors should consider in their effort to contribute meaningfully to the published literature. I appreciated the opportunity to read and review.

Major Concerns

1. The central focus of this work – namely, trade-offs between stability and metabolic energy cost – are immediately presumed from the very first sentence to exist in human locomotion. I recommend introducing this notion more thoughtfully with a broad readership in mind. In addition, that individuals weigh these tradeoff decisions consciously and purposefully is a central tenant of the manuscript as current written. Providing convincing evidence of this throughout is paramount.

2. In my opinion, the results tend not to convincingly support the eventual conclusions in the manuscript, most notably those represented in the title. The only outcome that appears to support this conclusion (that instability but not metabolic cost increased significantly during Amplification Low) is one the authors are cautious to mention is underpowered in the present manuscript. Rather than exhibiting a fundamentally different response, the increase in metabolic cost from Null to Low to High, on average, appears rather linear and may simply have failed to reach significance with the number of subjects in this manuscript. Particularly with the authors' own caveat in the results narrative (which I fully appreciate), I recommend the authors temper their interpretations and conclusions and consider an alternative title that avoids potentially misrepresenting the study outcomes. In addition, while I fully support thoughtful speculation in the discussion, I recommend the abstract rely less on this, and be revised to present a more objective summary of results.

3. Similarly, the results narrative too frequently includes subjective interpretation which should in all cases be relegated to the discussion. Generally, I think there is a gray area between evidence-based objective outcomes and the author's interpretations that is pervasive throughout the manuscript. One solution is to more explicitly disclose when something is your interpretation of the data versus something objectively observed.

Minor Concerns

Abstract

L6 "High energy expenditure" is relative – usual walking in otherwise health people is often considered relatively inexpensive. Here and in the introduction, this requires some context.

Introduction

P4 L17. Consider revising: "To maintain a straight forward walking trajectory, on average, the..."

P6 L50. Evidence for mechanism driving the increase in metabolic energy cost reported in O'Connor (i.e., variability) appears to be simple correlations. However, as the data in this

manuscript show, the mechanistic link between variability and metabolic cost may be more complex and thus not easily established.

P8 L6 and L21. Rather than describe in generalities (e.g., gait kinematics + CoM dynamics), I recommend the authors be more specific in their reference to outcome measures used to test their hypotheses.

Methods

P9 L30-48. The rationale for the inclusion of this visual feedback is not obvious until later in the methods narrative. This was a bit confusing. Consider reorganizing to make this clearer.

P12 L24. Please provide some evidence, perhaps from the authors prior studies, that 2 min was sufficient to wash our residual effects of the force field.

P12 L37. How often does the force field saturate at this 80 N threshold? This is particularly relevant for the Amplification High field, which was deemed the largest possible gain achievable. If frequently saturated, how does this influence the study outcomes?

P13 L30. The methodological decision to exclude arm swing could influence the trade-offs at the heart of this experiment and the ecological validity of the study outcomes. Please disclose as a limitation and thoughtfully discuss the implications.

P14 L24. Although only personal preference, I recommend the authors adopt the phrasing “work performed” over “work done” in all cases throughout the manuscript.

P15 L8-22. The authors verify the efficacy of the force field to amplify CoM motion not by analyzing CoM motion, but by analyzing work performed on the individual. The figures themselves show relatively little effect on the amplification of CoM movement during late exposure. Perhaps reconsider the phrasing here – what is it specifically that you intend to verify regarding system efficacy?

P15 L27. I would rather the authors justify their state space variables based on their functional relevance to balance rather than on a specific attribute of their force field controller.

P16 L8. There is an inconsistency in this paragraph; as the authors disclose, this correlation can allude to control strategies or passive dynamics. Thus, the motivation provided in the topic sentence (underlying control strategies only) is incomplete. Consider rephrasing.

P17 L24. Please provide the Bonferonni-adjusted level of significance used for the pairwise comparisons (only the unadjusted alpha level of 0.05 is provided).

Results

P17 L55. Revise: “mass”

I recommend the authors considering including effect sizes for all statistical comparisons.

Figure 6. Please consider reorganizing the panels to consolidate like outcomes and aid the reader in digesting all of this material (e.g., step length and width, step width and length variability).

Alternatively, the authors may consider separating center of mass from step placement outcomes in two figures.

Discussion

P24 L20. The authors should be more specific here about which adaptations specifically support the efficacy of this paradigm to probe trade-offs between stability and energy expenditure.

Biomechanical outcomes certainly changed in the presence of the force field, but is any change therein sufficient to provide support for this claim?

P26 L3-12. The subheading and the topic sentence here are a bit misleading, in my opinion. It is only the authors' interpretation of the objective outcomes that includes the possibility of different mechanisms at work. In reality, we don't really know the extent to which any of these mechanisms is actually being used, nor the extent to which their relative influence may change with force field magnitude.

P27 L40. The need for interventions designed to address instability are obvious to me. The need for those to address tradeoffs between instability and metabolic energy cost are much less

obvious. I recommend better clarifying the need for such intervention in this last sentence, as it seems central to the impact of this work and to the authors' research moving forward. P28 L21. More out of my own personal curiosity, why does the predictability of this paradigm not at least in part preclude its application to training and rehabilitation? Is there a specific therapeutic opportunity in mind that is fundamentally different from what one might target with random perturbations?

Review form: Reviewer 2

Is the manuscript scientifically sound in its present form?

No

Are the interpretations and conclusions justified by the results?

No

Is the language acceptable?

No

Do you have any ethical concerns with this paper?

No

Have you any concerns about statistical analyses in this paper?

Yes

Recommendation?

Major revision is needed (please make suggestions in comments)

Comments to the Author(s)

Overall, this is a nicely conducted study that has been badly buried in excessive analyses and a rather incoherently structured manuscript. The experiment itself is rather elegant. But the manuscript and analyses lack any real clear focus and so any elegance gets lost in the "noise". On the plus side, I think with minimal additional analyses (mostly just some slightly different stats, etc) and with some major re-writing and re-focusing of the narrative of the manuscript itself, this work has potential to be quite a nice paper.

I highly recommend that the authors - before they start to revise the manuscript - (and perhaps before they even read my detailed comments below) - read the "Ten Simple Rules for Structuring Papers" article by Mensch & Kording, 2017:
<https://doi.org/10.1371/journal.pcbi.1005619>

Detailed Comments as Follows:

ABSTRACT (p. 3):

The abstract in general is a bit vague - Would be stronger if re-written somewhat to be more specific. For example - 2nd sentence mentions "destabilization" and "uncertainty" - Are these the same thing? - Different things? what is meant by each of these in this specific context? Readers who have not read the paper yet will not know.

Another example: L. ~26-27: "When destabilization was less challenging..." - Readers who have not read the paper yet will have no idea what this means.

Also, 1st sentence of 2nd paragraph (p. 3, L. ~24-26) seems to present the "conclusion" before reporting the actual "result". Likewise, 2nd sentence ("... participants *may have chosen*...") - This is an "interpretation" again presented before that actual "result".

INTRODUCTION:

Provide Context?: P. 4, 1st paragraph jumps right in and presents a lot of concepts. Readers who are already very familiar with these ideas might be able to follow along. Readers with broader, more general background (i.e., readers of RSOS) will not.

For example, the very first sentence mentions "high energy expenditure", "risk for loss of balance", "trade-offs between stability and metabolic cost" and "significant consequences" - But none of these things are defined - How they are related is not defined, etc.

Likewise, the premise of the very first sentence is that there are indeed "trade-offs between stability and metabolic cost" - But it is not clear this is a "given". Indeed, others (e.g., Art Kuo, Roger Kram, Max Donelan, etc.) have argued the opposite - that indeed conditions of minimum metabolic cost should equate to "most stable" and vice versa.

So in general, re-structuring the Intro to provide more context and to define more specifically the terms and parameters of this study - to a more general RSOS audience - would strengthen the paper.

P. 5, L. ~40-41: "They weight the consequences of..." - This statement (and others like it) refer to biomechanical findings - However, this phrasing clearly implies some level of cognitive ("conscious") "intent" on the part of the person doing the "weighing". Such inferences are speculative at best. Statements like this should be re-phrased to focus more clearly on what actual facts we actually know. If/as the authors wish to speculate further, that is OK, but statements should be phrased very clearly to indicate such.

P. 5, L. ~49-54: Three things. First, again, this distinction between "uncertainty" and "balance challenge" has not been clearly defined.

P. 5, L. ~49-54: Second, the sentence now also equates "stability" to "high mechanical impedance". No basis or scientific premise for this is given and this is not always correct. Indeed, most classical limit cycle systems (van der Pol oscillators, passive dynamic walkers, etc. - often used to model walking!) exhibit dynamic stability that has no relation to their "mechanical impedance".

P. 5, L. ~54-56: Third, the sentence then offers a definition of mechanical impedance as "the forces that..." This definition may be fine for (and indeed derives from) purely passive mechanical systems - But that is a VERY different thing than a human (or robot etc.) that has active sensing, control, and actuation and that can therefore generate all sorts of forces "in response to imposed motions" that again have nothing to do with the system's mechanical impedance.

P. 6, L. ~3-8: Again, the example presented here attributes control actions (increasing step width) to increasing mechanical impedance, which conflates again the contributions of passive mechanics and active control.

P. 6, L. 10 - "... uncertainty of external perturbations..." - This phrase again mixes (I think) at least 3 different concepts: "uncertainty", "perturbations" and (by reference to the prior paragraph) "stability". Each of these could potentially have multiple definitions depending on the system and/or context being studied - None of these have yet been clearly defined here.

P. 6, L. ~28: Now the concepts of "energetics" and "local stability" are introduced - and again no clear definitions given. "Local stability" for example is a very specific type of "stability" (i.e., it is distinct from "global stability", "parametric stability", etc.). There are also different types of "local stability" with different operational definitions for different systems and contexts.

Remainder if Intro in General: The themes expressed in the comments above continue - I don't think it is helpful for me to single out each and every one. In general, the Intro needs work. The work presented here is clearly trying to tackle a larger problem ("balance" maybe?) that involves multiple interacting sub-components ("uncertainty", "perturbations", "stability", "energetic cost", etc.). For readers to be able to discern the contribution this work will make to the scientific literature, these terms and concepts need to be clearly defined (as much as possible) and a more clear and focused description of how they are inter-related is needed.

For example, the current Abstract and Intro include lots of discussion of "uncertainty" - But the "movement amplification" force field used in this study, in the end, does not even address this question because it is, by construction, "predictable"....

INTRODUCTION / METHODS:

It is clear that the viscous force fields imposed here are "predictable" - as in, they *can* be computed / predicted, they are not random, etc. But it is not clear to me that just because they *can* be predicted (mathematically) that we human beings can indeed predict them - The long-standing work on viscous curl fields in reaching suggests that we can, at least in some contexts - But I think this deserves more careful consideration.

METHODS:

P. 13, L. ~3-17: It is not clear here what was done to confirm the findings. If anthropometrics "influence" the accelerations and "there were absolute differences" observes in those accelerations, then what if any analyses were done to determine if and/or the extent to which these observed "differences" could be attributed to anthropometrics (or other factors)?

P. 13, L. ~23-29: I don't understand this sentence. Just because participants were "asked" to control their COM does not mean they necessarily did as they were asked. This may just be a phrasing issue, but the experimental design cannot entirely "distinguish" what participants were / were not "able" to do from what they "chose" to do. Certainly, instructions and so forth can help reduce these potential confounds, but cannot get rid of them entirely.

P. 14, L. ~8-10: It is mentioned here and else where that "participants had reached steady state". Several of these analyses depend on this assumption. In the study description (P. 12, L. ~14-19) it was "assumed" participants had reached steady state in the second trial. Was this verified / validated after-the-fact? If so, how?

P. 14, L. 38 - "... data was...": This is a common mistake, but the word "data" is plural, not singular ("datum" is singular). This should be "... data were..." etc. [see e.g.,: <https://blog.apastyle.org/apastyle/2012/07/data-is-or-data-are.html>]. This should be fixed throughout the manuscript.

P. 14, L. ~44-47: Was there a specific algorithm used to identify IC and TO events? Several have been published - If so, please cite. If some custom algorithm was derived, please describe and also justify why previously published methods were not able to be used here.

P. 15, L. ~10-15: Calculation not clear. Integrating power to compute work is fine - normalizing by body weight is fine and typical - But normalizing by # of strides taken does not make sense. Net work done over what time interval? Assuming over each stride, then you would obtain net work *for that stride* - how many strides is irrelevant. If integrating over the entire trial (all 100 strides at once), then dividing by 100 might make sense - but this would give you only total net work done over the entire trial and no measures from which to assess stride-to-stride differences in net work...??.

P. 15, L. 26: What does "Based on dynamical systems theory..." mean? Is there a competing theory for non-dynamical systems? What would a non "dynamical" system be anyway?: i.e., a system that exhibits no dynamics? Perhaps a rock? So would that mean the alternative to "dynamical systems theory" is a theory that all things are just rocks?
[Again - just trying to point out that terminology and careful definitions of terms does matter - This phrase "dynamical systems theory" is thrown around all the time - but almost never defined]

P. 15, L. ~24-45: Several published papers have also now shown that these algorithms for computing $\langle \lambda \rangle$ are also sensitive to a number of choices & parameters used in the analyses: definition of state space, choice of time delay, etc etc. How were the algorithmic choices made here validated?

P. 15, L. ~46 - P. 16, L. ~20: As discussed above, the Intro discusses multiple concepts ("balance", "uncertainty", "perturbations", "stability", "energetic cost", etc.). These two paragraphs here now refer to "control" also - And again, this not clearly defined. First, the basic biomechanical measures (P. 15) are fine, but by themselves may or may not reflect "mechanisms used to implement COM control" - This is an inference or speculation that is being imposed by the authors on these dependent measures - But the measures themselves are not measures of "control" etc.

P. 16, L. ~8-20: Likewise, this paragraph starts with the statement "to examine underlying control strategies...", but the metric (R^2 for regression of pelvis state to foot placement) is a correlation - And it is well known that correlation may or may not imply causation. So the notion of this as a measure of "control" is again an inference (i.e., speculation) by the authors. Indeed, their last sentence states that this R^2 metric can reflect "either increased feedback control or effects of passive dynamics" - So it is clearly not a measure of "control" per se.

P. 16, L. 37-40: The statement that "Net work could not be calculated..." is wrong. As the cables were "slack", the force applied by them was, by construction, zero - Therefore the net work was, by definition, also zero. Was this not why you did the first set of t-tests (L. ~33) to determine if Net Work was different from zero?

Second - This statement is not related to "Statistical Analysis" so probably does not belong in this paragraph anyway.

P. 16., L. ~42-44: If Bonferroni corrections were used for 3 comparisons, then the significance level should not be $p < 0.05$, but $p < 0.05/3$, i.e., $p < 0.01667$.

P. 17, L. ~21-23: So Bonferroni corrections for the multiple comparisons between conditions for each measure is one issue... But there is a separate issue of Type I error related to the fact that

these analyses were run on a large number of dependent measures (at least 9 acc. to L. ~10-15). Thus, the appropriate significance level should not be $p < 0.05$, but $p < 0.05/9$, i.e., $p < 0.00555$.

RESULTS:

"Participants": This does not seem to belong in "Results". The participants themselves are not the "results" - They are the participants from whom these results were obtained. Move to "Methods".

Figure 4: What is not described in the text, and was not tested for statistically, but appear in the figure, are differences between Early and Late Exposure. In particular, it appears that from Early to Late, Net Work increased for the Low field, but decreased for the High field. A 2x2 ANOVA (i.e., Early/Late x Low/High) would be needed to confirm this. In particular, Fig. 4 looks like there might be a significant interaction effect here. If so, what would this mean? How would this be interpreted?

Figure 5: So the title of this paper and the main narrative are that there is some "trade-off" between "stability" and metabolic cost (here, COT). But this figure appears to contradict that interpretation. There is a clear trend of increasing instability across conditions in Fig. 5a [& Stats confirmed that all of these differences were "significant"]. There is also a fairly clear trend increasing COT across conditions in Fig. 5b. No p-value is given for the Null vs. Low comparison, but even if this difference was not "statistically significant", this is exceptionally weak evidence of a "trade-off" b/t stability and COT. Indeed, the Discussion (P. 25, L. ~5-10) concedes as much: i.e., "*If* the two conditions were in fact equivalent, then participants *may* have chosen to sacrifice some stability... to maintain metabolic cost" [emphasis added] - This is a huge amount of hedging, especially when the authors' own equivalence test specifically could *not* establish the "equivalence" that the authors' primary conclusion is based upon.

Instead, the overall result - looking at all of Fig. 5 together seems to far more clearly indicate the opposite of any such trade-off. Indeed, if data were pooled across conditions, and a direct correlation of $\langle \lambda_S \rangle$ to COT were performed, it seems highly likely this would yield a strong and *positive* (not negative) correlation.

Figure 6 & the Corresponding Results: This figure is particularly difficult to 'decipher' as it mixes multiple variables that relate to multiple underlying constructs and also presents them in a somewhat random / haphazard order [e.g., why is step width variability (on panel (f)) not next to step width (in panel (b)), etc. - or, e.g., it might make sense to plot step widths (b) next to step lengths (d), but they are not - and/or, why are the more derived measures (MOS in (c), R^2 in (g)) thrown in with the basic stepping parameters they were derived from?].

Similarly, it is hard to see how R^2 (g) is conceptually related to step length variability (h) is related to mean DST's (i) - Yet, these 3 variables are plotted side-by-side as though the authors are expecting readers to draw such comparisons (even if only visually / qualitatively). In general, there just does not seem to be much rhyme-or-reason to this figure - It is not at all clear what overriding "result" this figure and/or these data are intended to portray. It is not at all clear what "question" these results are intended to answer.

Overall, as currently presented, it seems the only real main "finding" is that when you expose people to these destabilizing lateral perturbations - they become more destabilized.

DISCUSSION:

Sub-Section 1 - "Trade-offs between energetics and stability...": As "uncertainty" was not introduced nor manipulated in this experiment (there were no "certain" vs. "uncertain" conditions

to compare), any discussions or conclusions about "uncertainty" seem out of place here, as they cannot be supported (either for or against) by any of the results presented.

Sub-Section 2 - "Differences in metabolic cost... may be explained by mechanical work": Again, this discussion, and any conclusions drawn from it are not really supported by the results, as currently presented. The authors have both Net Work data and metabolic cost (COT) data - So if how these are related is an important topic of this work, then direct correlations of these measures should be conducted to substantiate any claims regarding how they are related.

Sub-Section 3 - "Biomechanical mechanisms suggest...": This section describes "control methods, including high impedance, increased feedback control, and/or decreased volitional control" etc. However, these constructs do not seem related to the actual data presented. none of the measures calculated are measures of "impedance" or of "feedback control" or of "volitional control" etc.

Etc.

In General, the current Discussion covers many topics that stray quite far from the actual experiment that was implemented and the specific dependent measures computed and presented. As with the Introduction, what seems missing here is a clear (and preferably more linear) narrative (e.g., 'What is A? How does A lead to B? How then does B lead to C?', etc.).

The Discussion and the manuscript as a whole could be very much improved if all of this ancillary and not-directly-relevant (and highly speculative) "stuff" was removed and the paper re-written to focus much more specifically on exactly what were the main, important, significant questions asked? - And exactly how does each result presented speak specifically to each of those questions? Etc.

Decision letter (RSOS-190889.R0)

10-Jul-2019

Dear Ms Wu,

The editors assigned to your paper ("A novel Movement Amplification environment reveals stability-metabolic trade-offs dependent on destabilization magnitude") have now received comments from reviewers. We would like you to revise your paper in accordance with the referee and Associate Editor suggestions which can be found below (not including confidential reports to the Editor). Please note this decision does not guarantee eventual acceptance.

Please submit a copy of your revised paper before 02-Aug-2019. Please note that the revision deadline will expire at 00.00am on this date. If we do not hear from you within this time then it will be assumed that the paper has been withdrawn. In exceptional circumstances, extensions may be possible if agreed with the Editorial Office in advance. We do not allow multiple rounds of revision so we urge you to make every effort to fully address all of the comments at this stage. If deemed necessary by the Editors, your manuscript will be sent back to one or more of the original reviewers for assessment. If the original reviewers are not available, we may invite new reviewers.

To revise your manuscript, log into <http://mc.manuscriptcentral.com/rsos> and enter your

Author Centre, where you will find your manuscript title listed under "Manuscripts with Decisions." Under "Actions," click on "Create a Revision." Your manuscript number has been appended to denote a revision. Revise your manuscript and upload a new version through your Author Centre.

- Data accessibility

If you wish to submit your supporting data or code to Dryad (<http://datadryad.org/>), or modify your current submission to dryad, please use the following link:
<http://datadryad.org/submit?journalID=RSOS&manu=RSOS-190889>

- Competing interests

- Authors' contributions

AB carried out the molecular lab work, participated in data analysis, carried out sequence alignments, participated in the design of the study and drafted the manuscript; CD carried out the statistical analyses; EF collected field data; GH conceived of the study, designed the study,

coordinated the study and helped draft the manuscript. All authors gave final approval for publication.

- Acknowledgements

- Funding statement

Kind regards,

Alice Power

Editorial Coordinator

on behalf of Dr Derek Abbott (Associate Editor) and R. Kerry Rowe (Subject Editor)

Comments to Author:

Reviewers' Comments to Author:

Reviewer: 1

Comments to the Author(s)

This study uses a novel lateral force field to test for the presence of trade-offs between stability and metabolic energy cost during walking in the absence of costs one might expect due to anticipatory changes arising from a threat to balance. The experimental approach is innovative, and the outcomes are interesting and relevant to the field of biomechanics and motor control. My most significant concern (see below) is that the results tend not to convincingly support the conclusions. I outline below several opportunities the authors should consider in their effort to contribute meaningfully to the published literature. I appreciated the opportunity to read and review.

Major Concerns

1. The central focus of this work – namely, trade-offs between stability and metabolic energy cost – are immediately presumed from the very first sentence to exist in human locomotion. I recommend introducing this notion more thoughtfully with a broad readership in mind. In addition, that individuals weigh these tradeoff decisions consciously and purposefully is a central tenant of the manuscript as current written. Providing convincing evidence of this throughout is paramount.

2. In my opinion, the results tend not to convincingly support the eventual conclusions in the manuscript, most notably those represented in the title. The only outcome that appears to support this conclusion (that instability but not metabolic cost increased significantly during Amplification Low) is one the authors are cautious to mention is underpowered in the present manuscript. Rather than exhibiting a fundamentally different response, the increase in metabolic cost from Null to Low to High, on average, appears rather linear and may simply have failed to

reach significance with the number of subjects in this manuscript. Particularly with the authors' own caveat in the results narrative (which I fully appreciate), I recommend the authors temper their interpretations and conclusions and consider an alternative title that avoids potentially misrepresenting the study outcomes. In addition, while I fully support thoughtful speculation in the discussion, I recommend the abstract rely less on this, and be revised to present a more objective summary of results.

3. Similarly, the results narrative too frequently includes subjective interpretation which should in all cases be relegated to the discussion. Generally, I think there is a gray area between evidence-based objective outcomes and the author's interpretations that is pervasive throughout the manuscript. One solution is to more explicitly disclose when something is your interpretation of the data versus something objectively observed.

Minor Concerns

Abstract

L6 "High energy expenditure" is relative – usual walking in otherwise health people is often considered relatively inexpensive. Here and in the introduction, this requires some context.

Introduction

P4 L17. Consider revising: "To maintain a straight forward walking trajectory, on average, the..."

P6 L50. Evidence for mechanism driving the increase in metabolic energy cost reported in O'Connor (i.e., variability) appears to be simple correlations. However, as the data in this manuscript show, the mechanistic link between variability and metabolic cost may be more complex and thus not easily established.

P8 L6 and L21. Rather than describe in generalities (e.g., gait kinematics + CoM dynamics), I recommend the authors be more specific in their reference to outcome measures used to test their hypotheses.

Methods

P9 L30-48. The rationale for the inclusion of this visual feedback is not obvious until later in the methods narrative. This was a bit confusing. Consider reorganizing to make this clearer.

P12 L24. Please provide some evidence, perhaps from the authors prior studies, that 2 min was sufficient to wash our residual effects of the force field.

P12 L37. How often does the force field saturate at this 80 N threshold? This is particularly relevant for the Amplification High field, which was deemed the largest possible gain achievable. If frequently saturated, how does this influence the study outcomes?

P13 L30. The methodological decision to exclude arm swing could influence the trade-offs at the heart of this experiment and the ecological validity of the study outcomes. Please disclose as a limitation and thoughtfully discuss the implications.

P14 L24. Although only personal preference, I recommend the authors adopt the phrasing "work performed" over "work done" in all cases throughout the manuscript.

P15 L8-22. The authors verify the efficacy of the force field to amplify CoM motion not by analyzing CoM motion, but by analyzing work performed on the individual. The figures themselves show relatively little effect on the amplification of CoM movement during late exposure. Perhaps reconsider the phrasing here – what is it specifically that you intend to verify regarding system efficacy?

P15 L27. I would rather the authors justify their state space variables based on their functional relevance to balance rather than on a specific attribute of their force field controller.

P16 L8. There is an inconsistency in this paragraph; as the authors disclose, this correlation can allude to control strategies or passive dynamics. Thus, the motivation provided in the topic sentence (underlying control strategies only) is incomplete. Consider rephrasing.

P17 L24. Please provide the Bonferonni-adjusted level of significance used for the pairwise comparisons (only the unadjusted alpha level of 0.05 is provided).

Results

P17 L55. Revise: "mass"

I recommend the authors considering including effect sizes for all statistical comparisons.

Figure 6. Please consider reorganizing the panels to consolidate like outcomes and aid the reader in digesting all of this material (e.g., step length and width, step width and length variability).

Alternatively, the authors may consider separating center of mass from step placement outcomes in two figures.

Discussion

P24 L20. The authors should be more specific here about which adaptations specifically support the efficacy of this paradigm to probe trade-offs between stability and energy expenditure.

Biomechanical outcomes certainly changed in the presence of the force field, but is any change therein sufficient to provide support for this claim?

P26 L3-12. The subheading and the topic sentence here are a bit misleading, in my opinion. It is only the authors' interpretation of the objective outcomes that includes the possibility of different mechanisms at work. In reality, we don't really know the extent to which any of these mechanisms is actually being used, nor the extent to which their relative influence may change with force field magnitude.

P27 L40. The need for interventions designed to address instability are obvious to me. The need for those to address tradeoffs between instability and metabolic energy cost are much less obvious. I recommend better clarifying the need for such intervention in this last sentence, as it seems central to the impact of this work and to the authors' research moving forward.

P28 L21. More out of my own personal curiosity, why does the predictability of this paradigm not at least in part preclude its application to training and rehabilitation? Is there a specific therapeutic opportunity in mind that is fundamentally different from what one might target with random perturbations?

Reviewer: 2

Comments to the Author(s)

Overall, this is a nicely conducted study that has been badly buried in excessive analyses and a rather incoherently structured manuscript. The experiment itself is rather elegant. But the manuscript and analyses lack any real clear focus and so any elegance gets lost in the "noise". On the plus side, I think with minimal additional analyses (mostly just some slightly different stats, etc) and with some major re-writing and re-focusing of the narrative of the manuscript itself, this work has potential to be quite a nice paper.

I highly recommend that the authors - before they start to revise the manuscript - (and perhaps before they even read my detailed comments below) - read the "Ten Simple Rules for Structuring Papers" article by Mensch & Kording, 2017:

<https://doi.org/10.1371/journal.pcbi.1005619>

Detailed Comments as Follows:

ABSTRACT (p. 3):

The abstract in general is a bit vague - Would be stronger if re-written somewhat to be more specific. For example - 2nd sentence mentions "destabilization" and "uncertainty" - Are these the

same thing? - Different things? what is meant by each of these in this specific context? Readers who have not read the paper yet will not know.

Another example: L. ~26-27: "When destabilization was less challenging..." - Readers who have not read the paper yet will have no idea what this means.

Also, 1st sentence of 2nd paragraph (p. 3, L. ~24-26) seems to present the "conclusion" before reporting the actual "result". Likewise, 2nd sentence ("... participants *may have chosen*...") - This is an "interpretation" again presented before that actual "result".

INTRODUCTION:

Provide Context?: P. 4, 1st paragraph jumps right in and presents a lot of concepts. Readers who are already very familiar with these ideas might be able to follow along. Readers with broader, more general background (i.e., readers of RSOS) will not.

For example, the very first sentence mentions "high energy expenditure", "risk for loss of balance", "trade-offs between stability and metabolic cost" and "significant consequences" - But none of these things are defined - How they are related is not defined, etc.

Likewise, the premise of the very first sentence is that there are indeed "trade-offs between stability and metabolic cost" - But it is not clear this is a "given". Indeed, others (e.g., Art Kuo, Roger Kram, Max Donelan, etc.) have argued the opposite - that indeed conditions of minimum metabolic cost should equate to "most stable" and vice versa.

So in general, re-structuring the Intro to provide more context and to define more specifically the terms and parameters of this study - to a more general RSOS audience - would strengthen the paper.

P. 5, L. ~40-41: "They weight the consequences of..." - This statement (and others like it) refer to biomechanical findings - However, this phrasing clearly implies some level of cognitive ("conscious") "intent" on the part of the person doing the "weighing". Such inferences are speculative at best. Statements like this should be re-phrased to focus more clearly on what actual facts we actually know. If/as the authors wish to speculate further, that is OK, but statements should be phrased very clearly to indicate such.

P. 5, L. ~49-54: Three things. First, again, this distinction between "uncertainty" and "balance challenge" has not been clearly defined.

P. 5, L. ~49-54: Second, the sentence now also equates "stability" to "high mechanical impedance". No basis or scientific premise for this is given and this is not always correct. Indeed, most classical limit cycle systems (van der Pol oscillators, passive dynamic walkers, etc. - often used to model walking!) exhibit dynamic stability that has no relation to their "mechanical impedance".

P. 5, L. ~54-56: Third, the sentence then offers a definition of mechanical impedance as "the forces that..." This definition may be fine for (and indeed derives from) purely passive mechanical systems - But that is a VERY different thing than a human (or robot etc.) that has active sensing, control, and actuation and that can therefore generate all sorts of forces "in response to imposed motions" that again have nothing to do with the system's mechanical impedance.

P. 6, L. ~3-8: Again, the example presented here attributes control actions (increasing step width)

to increasing mechanical impedance, which conflates again the contributions of passive mechanics and active control.

P. 6, L. 10 - "... uncertainty of external perturbations..." - This phrase again mixes (I think) at least 3 different concepts: "uncertainty", "perturbations" and (by reference to the prior paragraph) "stability". Each of these could potentially have multiple definitions depending on the system and/or context being studied - None of these have yet been clearly defined here.

P. 6, L. ~28: Now the concepts of "energetics" and "local stability" are introduced - and again no clear definitions given. "Local stability" for example is a very specific type of "stability" (i.e., it is distinct from "global stability", "parametric stability", etc.). There are also different types of "local stability" with different operational definitions for different systems and contexts.

Remainder if Intro in General: The themes expressed in the comments above continue - I don't think it is helpful for me to single out each and every one. In general, the Intro needs work. The work presented here is clearly trying to tackle a larger problem ("balance" maybe?) that involves multiple interacting sub-components ("uncertainty", "perturbations", "stability", "energetic cost", etc.). For readers to be able to discern the contribution this work will make to the scientific literature, these terms and concepts need to be clearly defined (as much as possible) and a more clear and focused description of how they are inter-related is needed.

For example, the current Abstract and Intro include lots of discussion of "uncertainty" - But the "movement amplification" force field used in this study, in the end, does not even address this question because it is, by construction, "predictable"....

INTRODUCTION / METHODS:

It is clear that the viscous force fields imposed here are "predictable" - as in, they *can* be computed / predicted, they are not random, etc. But it is not clear to me that just because they *can* be predicted (mathematically) that we human beings can indeed predict them - The long-standing work on viscous curl fields in reaching suggests that we can, at least in some contexts - But I think this deserves more careful consideration.

METHODS:

P. 13, L. ~3-17: It is not clear here what was done to confirm the findings. If anthropometrics "influence" the accelerations and "there were absolute differences" observes in those accelerations, then what if any analyses were done to determine if and/or the extent to which these observed "differences" could be attributed to anthropometrics (or other factors)?

P. 13, L. ~23-29: I don't understand this sentence. Just because participants were "asked" to control their COM does not mean they necessarily did as they were asked. This may just be a phrasing issue, but the experimental design cannot entirely "distinguish" what participants were / were not "able" to do from what they "chose" to do. Certainly, instructions and so forth can help reduce these potential confounds, but cannot get rid of them entirely.

P. 14, L. ~8-10: It is mentioned here and else where that "participants had reached steady state". Several of these analyses depend on this assumption. In the study description (P. 12, L. ~14-19) it was "assumed" participants had reached steady state in the second trial. Was this verified / validated after-the-fact? If so, how?

P. 14, L. 38 - "... data was...": This is a common mistake, but the word "data" is plural, not singular ("datum" is singular). This should be "... data were..." etc. [see e.g.,:

<https://blog.apastyle.org/apastyle/2012/07/data-is-or-data-are.html>]. This should be fixed throughout the manuscript.

P. 14, L. ~44-47: Was there a specific algorithm used to identify IC and TO events? Several have been published - If so, please cite. If some custom algorithm was derived, please describe and also justify why previously published methods were not able to be used here.

P. 15, L. ~10-15: Calculation not clear. Integrating power to compute work is fine - normalizing by body weight is fine and typical - But normalizing by # of strides taken does not make sense. Net work done over what time interval? Assuming over each stride, then you would obtain net work *for that stride* - how many strides is irrelevant. If integrating over the entire trial (all 100 strides at once), then dividing by 100 might make sense - but this would give you only total net work done over the entire trial and no measures from which to assess stride-to-stride differences in net work...??.

P. 15, L. 26: What does "Based on dynamical systems theory..." mean? Is there a competing theory for non-dynamical systems? What would a non "dynamical" system be anyway?: i.e., a system that exhibits no dynamics? Perhaps a rock? So would that mean the alternative to "dynamical systems theory" is a theory that all things are just rocks?
[Again - just trying to point out that terminology and careful definitions of terms does matter - This phrase "dynamical systems theory" is thrown around all the time - but almost never defined]

P. 15, L. ~24-45: Several published papers have also now shown that these algorithms for computing $\langle \lambda \rangle$ are also sensitive to a number of choices & parameters used in the analyses: definition of state space, choice of time delay, etc etc. How were the algorithmic choices made here validated?

P. 15, L. ~46 - P. 16, L. ~20: As discussed above, the Intro discusses multiple concepts ("balance", "uncertainty", "perturbations", "stability", "energetic cost", etc.). These two paragraphs here now refer to "control" also - And again, this not clearly defined. First, the basic biomechanical measures (P. 15) are fine, but by themselves may or may not reflect "mechanisms used to implement COM control" - This is an inference or speculation that is being imposed by the authors on these dependent measures - But the measures themselves are not measures of "control" etc.

P. 16, L. ~8-20: Likewise, this paragraph starts with the statement "to examine underlying control strategies...", but the metric (R^2 for regression of pelvis state to foot placement) is a correlation - And it is well known that correlation may or may not imply causation. So the notion of this as a measure of "control" is again an inference (i.e., speculation) by the authors. Indeed, their last sentence states that this R^2 metric can reflect "either increased feedback control or effects of passive dynamics" - So it is clearly not a measure of "control" per se.

P. 16, L. 37-40: The statement that "Net work could not be calculated..." is wrong. As the cables were "slack", the force applied by them was, by construction, zero - Therefore the net work was, by definition, also zero. Was this not why you did the first set of t-tests (L. ~33) to determine if Net Work was different from zero?

Second - This statement is not related to "Statistical Analysis" so probably does not belong in this paragraph anyway.

P. 16., L. ~42-44: If Bonferroni corrections were used for 3 comparisons, then the significance level should not be $p < 0.05$, but $p < 0.05/3$, i.e., $p < 0.01667$.

P. 17, L. ~21-23: So Bonferroni corrections for the multiple comparisons between conditions for each measure is one issue... But there is a separate issue of Type I error related to the fact that these analyses were run on a large number of dependent measures (at least 9 acc. to L. ~10-15). Thus, the appropriate significance level should not be $p < 0.05$, but $p < 0.05/9$, i.e., $p < 0.00555$.

RESULTS:

"Participants": This does not seem to belong in "Results". The participants themselves are not the "results" - They are the participants from whom these results were obtained. Move to "Methods".

Figure 4: What is not described in the text, and was not tested for statistically, but appear in the figure, are differences between Early and Late Exposure. In particular, it appears that from Early to Late, Net Work increased for the Low field, but decreased for the High field. A 2x2 ANOVA (i.e., Early/Late x Low/High) would be needed to confirm this. In particular, Fig. 4 looks like there might be a significant interaction effect here. If so, what would this mean? How would this be interpreted?

Figure 5: So the title of this paper and the main narrative are that there is some "trade-off" between "stability" and metabolic cost (here, COT). But this figure appears to contradict that interpretation. There is a clear trend of increasing instability across conditions in Fig. 5a [& Stats confirmed that all of these differences were "significant"]. There is also a fairly clear trend increasing COT across conditions in Fig. 5b. No p-value is given for the Null vs. Low comparison, but even if this difference was not "statistically significant", this is exceptionally weak evidence of a "trade-off" b/t stability and COT. Indeed, the Discussion (P. 25, L. ~5-10) concedes as much: i.e., "If* the two conditions were in fact equivalent, then participants *may* have chosen to sacrifice some stability... to maintain metabolic cost" [emphasis added] - This is a huge amount of hedging, especially when the authors' own equivalence test specifically could *not* establish the "equivalence" that the authors' primary conclusion is based upon.

Instead, the overall result - looking at all of Fig. 5 together seems to far more clearly indicate the opposite of any such trade-off. Indeed, if data were pooled across conditions, and a direct correlation of $\langle \lambda_S \rangle$ to COT were performed, it seems highly likely this would yield a strong and *positive* (not negative) correlation.

Figure 6 & the Corresponding Results: This figure is particularly difficult to 'decipher' as it mixes multiple variables that relate to multiple underlying constructs and also presents them in a somewhat random / haphazard order [e.g., why is step width variability (on panel (f)) not next to step width (in panel (b)), etc. - or, e.g., it might make sense to plot step widths (b) next to step lengths (d), but they are not - and/or, why are the more derived measures (MOS in (c), R^2 in (g)) thrown in with the basic stepping parameters they were derived from?].

Similarly, it is hard to see how R^2 (g) is conceptually related to step length variability (h) is related to mean DST's (i) - Yet, these 3 variables are plotted side-by-side as though the authors are expecting readers to draw such comparisons (even if only visually / qualitatively). In general, there just does not seem to be much rhyme-or-reason to this figure - It is not at all clear what overriding "result" this figure and/or these data are intended to portray. It is not at all clear what "question" these results are intended to answer.

Overall, as currently presented, it seems the only real main "finding" is that when you expose people to these destabilizing lateral perturbations - they become more destabilized.

DISCUSSION:

Sub-Section 1 - "Trade-offs between energetics and stability...": As "uncertainty" was not introduced nor manipulated in this experiment (there were no "certain" vs. "uncertain" conditions to compare), any discussions or conclusions about "uncertainty" seem out of place here, as they cannot be supported (either for or against) by any of the results presented.

Sub-Section 2 - "Differences in metabolic cost... may be explained by mechanical work": Again, this discussion, and any conclusions drawn from it are not really supported by the results, as currently presented. The authors have both Net Work data and metabolic cost (COT) data - So if how these are related is an important topic of this work, then direct correlations of these measures should be conducted to substantiate any claims regarding how they are related.

Sub-Section 3 - "Biomechanical mechanisms suggest...": This section describes "control methods, including high impedance, increased feedback control, and/or decreased volitional control" etc. However, these constructs do not seem related to the actual data presented. none of the measures calculated are measures of "impedance" or of "feedback control" or of "volitional control" etc.

Etc.

In General, the current Discussion covers many topics that stray quite far from the actual experiment that was implemented and the specific dependent measures computed and presented. As with the Introduction, what seems missing here is a clear (and preferably more linear) narrative (e.g., 'What is A? How does A lead to B? How then does B lead to C?', etc.).

The Discussion and the manuscript as a whole could be very much improved if all of this ancillary and not-directly-relevant (and highly speculative) "stuff" was removed and the paper re-written to focus much more specifically on exactly what were the main, important, significant questions asked? - And exactly how does each result presented speak specifically to each of those questions? Etc.

Author's Response to Decision Letter for (RSOS-190889.R0)

See Appendix A.

Decision letter (RSOS-190889.R1)

06-Nov-2019

Dear Ms Wu,

I am pleased to inform you that your manuscript entitled "A novel Movement Amplification environment reveals effects of controlling lateral centre of mass motion on gait stability and metabolic cost" is now accepted for publication in Royal Society Open Science.

Please note that the email address 'jwoodward@sralab.org' is not currently receiving messages - we need to ensure that this author has an alternative email address. Please can you let the editorial office know a better email at the address below?

on behalf of Dr Derek Abbott (Associate Editor) and R. Kerry Rowe (Subject Editor)
openscience@royalsociety.org

Appendix A

Reviewer: 1

Comments to the Author(s)

This study uses a novel lateral force field to test for the presence of trade-offs between stability and metabolic energy cost during walking in the absence of costs one might expect due to anticipatory changes arising from a threat to balance. The experimental approach is innovative, and the outcomes are interesting and relevant to the field of biomechanics and motor control. My most significant concern (see below) is that the results tend not to convincingly support the conclusions. I outline below several opportunities the authors should consider in their effort to contribute meaningfully to the published literature. I appreciated the opportunity to read and review.

We thank the reviewer for their careful reading of the manuscript and their thorough suggestions for revisions. We have substantially revised the manuscript. Their comments have been valuable for improving the quality of the manuscript. We have focused the manuscript on the result and outcome we observed, changes in foot placement and metabolic cost. We have made efforts to remove all speculative statements and have eliminated the prior focus on stability-energetic trade-offs.

Major Concerns

1. The central focus of this work – namely, trade-offs between stability and metabolic energy cost – are immediately presumed from the very first sentence to exist in human locomotion. I recommend introducing this notion more thoughtfully with a broad readership in mind. In addition, that individuals weigh these tradeoff decisions consciously and purposefully is a central tenant of the manuscript as current written. Providing convincing evidence of this throughout is paramount.

We agree that the original manuscript emphasized stability-energetic trade-offs from the beginning and this detracted from other concepts with stronger backing from our results. We have made major revisions and have changed the focus to understanding how people control lateral COM motion and the metabolic costs associated with these strategies.

2. In my opinion, the results tend not to convincingly support the eventual conclusions in the manuscript, most notably those represented in the title. The only outcome that appears to support this conclusion (that instability but not metabolic cost increased significantly during Amplification Low) is one the authors are cautious to mention is underpowered in the present manuscript. Rather than exhibiting a fundamentally different response, the increase in metabolic cost from Null to Low to High, on average, appears rather linear and may simply have failed to reach significance with the number of subjects in this manuscript. Particularly with the authors' own caveat in the results narrative (which I fully appreciate), I recommend the authors temper their interpretations and conclusions and consider an alternative title that avoids potentially misrepresenting the study outcomes. In addition, while I fully support thoughtful speculation in the discussion, I recommend the abstract rely less on this, and be revised to present a more objective summary of results.

After reviewing the comments and our results further, we agree that our data did not support the focus of the original manuscript, which was too speculative. We have made major revisions to emphasize the novel method of challenging control of lateral motion and to discuss its impact on gait stability, metabolic cost, and kinematics. We believe these results build on previous research that describing how people control the natural lateral oscillations of the body center of mass during gait.

We have thus edited the manuscript title, Abstract, Introduction, and Discussion to support this new focus. We have also edited the caption for Figure 5 (now figure 4) to be less speculative about equivalence between Null and Amplification Low fields for the COT data.

3. Similarly, the results narrative too frequently includes subjective interpretation which should in all cases be relegated to the discussion. Generally, I think there is a gray area between evidence-based objective outcomes and the author's interpretations that is pervasive throughout the manuscript. One solution is to more explicitly disclose when something is your interpretation of the data versus something objectively observed.

We agree that the original Results section had many speculative statements and have edited to present the data without interpretation, which is relegated to the Discussion. We have edited phrasing, where appropriate, to more explicitly disclose when something is an interpretation of the data by using terms such as "we believe."

Minor Concerns

Abstract

L6 "High energy expenditure" is relative – usual walking in otherwise healthy people is often considered relatively inexpensive. Here and in the introduction, this requires some context.

This statement was removed from the revised manuscript.

Introduction

P4 L17. Consider revising: "To maintain a straight forward walking trajectory, on average, the..."

This statement was removed from the revised manuscript.

P6 L50. Evidence for mechanism driving the increase in metabolic energy cost reported in O'Connor (i.e., variability) appears to be simple correlations. However, as the data in this manuscript show, the mechanistic link between variability and metabolic cost may be more complex and thus not easily established.

In refocusing the manuscript, we have removed reference to this article.

P8 L6 and L21. Rather than describe in generalities (e.g., gait kinematics + CoM dynamics), I recommend the authors be more specific in their reference to outcome measures used to test their hypotheses.

We have revised the last paragraph of the Introduction to specify our outcome measures we use to test our hypothesis P4 L24- P5 3:

"We hypothesize that compared to normal walking, when people are in the Movement Amplification field they will modulate their stepping patterns taking wider steps to aid in control of the lateral COM motion and that the strategies to control lateral motion will increase the metabolic cost of transport."

Methods

P9 L30-48. The rationale for the inclusion of this visual feedback is not obvious until later in the methods narrative. This was a bit confusing. Consider reorganizing to make this clearer.

We have reorganized the Setup and Methods to be one combined section and moved up the rationale for the visual feedback to earlier in the Methods section.

P12 L24. Please provide some evidence, perhaps from the authors prior studies, that 2 min was sufficient to wash out residual effects of the force field.

This is our first publication on this particular type of force field, and we chose a long washout period to be conservative. 2 minutes is much longer than the time necessary to complete 14 steps, which was the mean time constant we observed for participants to return to steady-state after exposure to a damping field in previous experiments (Wu et al. 2017). In general, we did not observe any aftereffects of the Movement Amplification field, and participants did not report perceptions of aftereffects. We have included the following details on P7 L14-16 of the revision: "...participants walked for an additional 2 minutes to wash out any residual effects of the force field, based on previous experiments showing aftereffects in a damping field (30)."

P12 L37. How often does the force field saturate at this 80 N threshold? This is particularly relevant for the Amplification High field, which was deemed the largest possible gain achievable. If frequently saturated, how does this influence the study outcomes?

The force field never saturated at 80N, even for the high magnitude Movement Amplification field. With the gain of the high field at 50 Ns/m, the lateral COM velocity would need to exceed 1.6 m/s, which did not occur in the experiments. Figure 5a in the revised manuscript shows that peak COM speeds did not exceed 0.25 m/s.

P13 L30. The methodological decision to exclude arm swing could influence the trade-offs at the heart of this experiment and the ecological validity of the study outcomes. Please disclose as a limitation and thoughtfully discuss the implications.

We chose to exclude arm swing in order to make the effect systematic across participants as explained in P9 L16 to P10 L1 of the revision:

"Not allowing arm swing has been shown to increase metabolic cost of walking (32-34), so we chose to include this restriction to make the effect consistent across participants as opposed to allowing participants to choose whether or not to use arm swing."

If we allowed participants to choose whether or not to swing their arms, this might have conflated our results such that we could not isolate what metabolic differences were due to the force field versus arm swing. We now identify the possible limitation of this method in the manuscript but we do believe this is an appropriate method for controlling the protocol.

P14 L24. Although only personal preference, I recommend the authors adopt the phrasing "work performed" over "work done" in all cases throughout the manuscript.

We have changed the phrasing to "work performed."

P15 L8-22. The authors verify the efficacy of the force field to amplify CoM motion not by analyzing CoM motion, but by analyzing work performed on the individual. The figures themselves show relatively little effect on the amplification of CoM movement during late exposure. Perhaps reconsider the phrasing here – what is it specifically that you intend to verify regarding system efficacy?

As stated on P14 L4-6, we wished to verify that the forces were in the same direction as the person's COM velocity:

“These results, along with visual observation (Video 1 at (<https://osf.io/5j7rk/>) and Figure 2), confirmed that the Amplification fields operated as intended during Early Exposure, pulling the person’s body in the direction of their movement”

P15 L27. I would rather the authors justify their state space variables based on their functional relevance to balance rather than on a specific attribute of their force field controller.

The reviewer is correct that, apart from the mediolateral velocity being a specific attribute of our force field controller, there is another reason why we chose to analyse mediolateral velocity. This is because there are several studies showing that walking is unstable in the mediolateral direction, and thus, it makes sense to investigate this direction. Moreover, we used velocity instead of position, in order to prevent nonstationarities in the data. This has now been described in more detail in the text on P11 L17-19: “We chose lateral COM velocity since walking is passively unstable in the mediolateral direction (1-3) and velocity is not affected by nonstationarities, as opposed to position data.”

P16 L8. There is an inconsistency in this paragraph; as the authors disclose, this correlation can allude to control strategies or passive dynamics. Thus, the motivation provided in the topic sentence (underlying control strategies only) is incomplete. Consider rephrasing.

We have removed the phrase “underlying control strategies” but refer to “gait strategies” in the topic sentence of the preceding paragraph (P12 L4). We believe that the last sentence of this paragraph adequately explains the interpretation of this metric: “The R^2 value of the regression fit was a metric of the degree of coupling between foot placement and COM dynamics, with higher correlation suggesting either increased neural control or effects of passive dynamics (10).” (P12 L14-16).

P17 L24. Please provide the Bonferroni-adjusted level of significance used for the pairwise comparisons (only the unadjusted alpha level of 0.05 is provided).

To clarify the procedure for Bonferroni adjustments, which were performed throughout, the new sentence reads “Bonferroni corrections were used for these three pairwise comparisons, and significance was set at the $p < 0.05$ level (i.e. the unadjusted pairwise comparison p-values were multiplied by 3 and then compared vs. 0.05).” on P13 L2-4. Thus we have already accounted for Bonferroni adjustments in our calculations.

Results

P17 L55. Revise: “mass”

We have changed the term “weight” to “body mass” throughout the revision.

I recommend the authors considering including effect sizes for all statistical comparisons.

We have omitted effect sizes in the revised manuscript, but will provide repository data necessary for readers to calculate these if desired.

Figure 6. Please consider reorganizing the panels to consolidate like outcomes and aid the reader in digesting all of this material (e.g., step length and width, step width and length variability). Alternatively, the authors may consider separating center of mass from step placement outcomes in two figures.

We have separated the kinematics-based results into two sections: one on mean values (Figure 5), and one on variability and degree of coupling between COM dynamics and foot placement (Figure 6).

Discussion

P24 L20. The authors should be more specific here about which adaptations specifically support the efficacy of this paradigm to probe trade-offs between stability and energy expenditure. Biomechanical outcomes certainly changed in the presence of the force field, but is any change therein sufficient to provide support for this claim?

We believe that the only result from the current study that may have ramifications for trade-offs between stability and metabolic cost is the non-significant difference in COT between Null and Amplification Low. Given that we could not verify equivalence, we have decided to minimize all references to the concept of trade-offs in stability and metabolic cost in this revision. The current Discussion focuses on describing the observed changes in foot placement and metabolic costs when walking in the Movement Amplification fields.

P26 L3-12. The subheading and the topic sentence here are a bit misleading, in my opinion. It is only the authors' interpretation of the objective outcomes that includes the possibility of different mechanisms at work. In reality, we don't really know the extent to which any of these mechanisms is actually being used, nor the extent to which their relative influence may change with force field magnitude.

We have used the terms "gait patterns," "gait strategies," and "gait kinematics" to refer to the data presented here in hopes of not overstating neural mechanisms. We agree that we have not measured cognitive intent. We have modified the subheading to "Changes in foot placement and other gait patterns suggest a combination of control strategies were employed in the Movement Amplification fields" and edited the topic sentence of the third paragraph to hopefully make clear that this is our interpretation of the results: "It is likely that these changes in step width were used in coordination with other control strategies." We feel that while this section may seem speculative, the attempt to relate kinematic data to motor control strategies is very important for interpreting results in gait stability and metabolic cost.

P27 L40. The need for interventions designed to address instability are obvious to me. The need for those to address tradeoffs between instability and metabolic energy cost are much less obvious. I recommend better clarifying the need for such intervention in this last sentence, as it seems central to the impact of this work and to the authors' research moving forward.

We have added a section to the discussion titled "Movement Amplification for Locomotor Training" where we discuss the potential clinical applications of this study. We have removed references to trade-offs between stability and metabolic cost.

P28 L21. More out of my own personal curiosity, why does the predictability of this paradigm not at least in part preclude its application to training and rehabilitation? Is there a specific therapeutic opportunity in mind that is fundamentally different from what one might target with random perturbations?

Thank you for your interest in this topic. We believe others will be interested as well. Thus, as mentioned in the previous comment, we have added a section to the discussion that presents why the Movement Amplification paradigm may have unique applications for training and rehabilitation.

Reviewer: 2

Comments to the Author(s)

Overall, this is a nicely conducted study that has been badly buried in excessive analyses and a rather incoherently structured manuscript. The experiment itself is rather elegant. But the manuscript and analyses lack any real clear focus and so any elegance gets lost in the "noise". On the plus side, I think with minimal additional analyses (mostly just some slightly different stats, etc) and with some major re-writing and re-focusing of the narrative of the manuscript itself, this work has potential to be quite a nice paper.

I highly recommend that the authors - before they start to revise the manuscript - (and perhaps before they even read my detailed comments below) - read the "Ten Simple Rules for Structuring Papers" article by Mensch & Kording, 2017:

<https://doi.org/10.1371/journal.pcbi.1005619>

We thank the reviewer for these comments and have restructured the paper and edited down the narrative to focus on the measurable outcomes from this study. We are familiar with the Mensch & Kording 2017 paper and have reviewed it.

Detailed Comments as Follows:

ABSTRACT (p. 3):

The abstract in general is a bit vague - Would be stronger if re-written somewhat to be more specific. For example - 2nd sentence mentions "destabilization" and "uncertainty" - Are these the same thing? - Different things? what is meant by each of these in this specific context? Readers who have not read the paper yet will not know.

We have substantially rewritten the Abstract. We believe the revised abstract is more clear and focused. We have made considerable effort to eliminate the use of vague and undefined terms in the abstract and throughout the manuscript. We have removed all usages of "destabilization" and "uncertainty."

Another example: L. ~26-27: "When destabilization was less challenging..." - Readers who have not read the paper yet will have no idea what this means.

We have removed this sentence and significantly edited the Abstract.

Also, 1st sentence of 2nd paragraph (p. 3, L. ~24-26) seems to present the "conclusion" before reporting the actual "result". Likewise, 2nd sentence ("... participants *may have chosen*...") - This is an "interpretation" again presented before that actual "result".

We have significantly edited the Abstract and addressed this concern.

INTRODUCTION:

Provide Context?: P. 4, 1st paragraph jumps right in and presents a lot of concepts. Readers who are already very familiar with these ideas might be able to follow along. Readers with broader, more general background (i.e., readers of RSOS) will not.

For example, the very first sentence mentions "high energy expenditure", "risk for loss of balance", "trade-offs between stability and metabolic cost" and "significant consequences" - But none of these things are defined - How they are related is not defined, etc.

Thank you for identifying these issues. We have substantially revised the introduction. The revised introduction has been written to provide a broad general background for the readers of RSOS. We have made efforts to remove vague and undefined concepts throughout the manuscript. The specific terms mentioned in this comment have been removed. In addition we have eliminated all references to trade-offs between stability and metabolic cost from the Introduction as this is no longer the focus of the manuscript.

Likewise, the premise of the very first sentence is that there are indeed "trade-offs between stability and metabolic cost" - But it is not clear this is a "given". Indeed, others (e.g., Art Kuo, Roger Kram, Max Donelan, etc.) have argued the opposite - that indeed conditions of minimum metabolic cost should equate to "most stable" and vice versa.

We agree that the original manuscript emphasized stability-energetic trade-offs from the beginning and detracted from other concepts with stronger backing from our results. We have made major revisions to shift the focus to understanding the methods people use to control lateral center of mass motion during walking and the associated metabolic energy costs. As mentioned earlier, the manuscript no longer focuses on trade-offs between stability and energy.

So in general, re-structuring the Intro to provide more context and to define more specifically the terms and parameters of this study - to a more general RSOS audience - would strengthen the paper.

We have substantially rewritten the manuscript, including the Introduction. The introduction has been written for a broader audience. We thank the reviewer for their comments and ideas on how to improve the presentation of our study.

P. 5, L. ~40-41: "They weight the consequences of..." - This statement (and others like it) refer to biomechanical findings - However, this phrasing clearly implies some level of cognitive ("conscious"?) "intent" on the part of the person doing the "weighing". Such inferences are speculative at best. Statements like this should be re-phrased to focus more clearly on what actual facts we actually know. If/as the authors wish to speculate further, that is OK, but statements should be phrased very clearly to indicate such.

We have rephrased the statement and similar ones to focus on the observed behavior rather than the person's cognitive intent. We agree that the term "weighing" implies conscious intent and have removed it. We have used phrases such as "it is likely" to highlight instances when we are interpreting or speculating on the results throughout the revised manuscript.

P. 5, L. ~49-54: Three things. First, again, this distinction between "uncertainty" and "balance challenge" has not been clearly defined.

We have removed all usages of the word "uncertainty."

We have removed usages of the term "balance challenge" from the Introduction and most of the manuscript to avoid confusion. We now consistently refer to the "magnitude" of amplification. We do refer to "challenge level" in terms of the visual feedback task alone (without external force field) and explain the measurement of challenge level in the Methods section (P6 L11). We refer several times to

the “challenge of controlling COM motion,” which we believe is justified by the design of the Movement Amplification force field.

P. 5, L. ~49-54: Second, the sentence now also equates "stability" to "high mechanical impedance". No basis or scientific premise for this is given and this is not always correct. Indeed, most classical limit cycle systems (van der Pol oscillators, passive dynamic walkers, etc. - often used to model walking!) exhibit dynamic stability that has no relation to their "mechanical impedance".

We have significantly edited the manuscript to remove all references to the concept of mechanical impedance, including the sentence referred to by the reviewer. We felt that this concept detracted from the various other concepts already presented.

P. 5, L. ~54-56: Third, the sentence then offers a definition of mechanical impedance as "the forces that..." This definition may be fine for (and indeed derives from) purely passive mechanical systems - But that is a VERY different thing than a human (or robot etc.) that has active sensing, control, and actuation and that can therefore generate all sorts of forces "in response to imposed motions" that again have nothing to do with the system's mechanical impedance.

We also felt that we should not discuss mechanical impedance since we did not directly measure it in this study. We agree that a purely passive mechanical system is very different than a human or robot with active sensing, control, and actuation, but we also believe that measurements of COM impedance in the future would significantly aid understanding of complex combinations of gait patterns in non-impaired walking as well as walking in persons with impaired sensing, control, and/or actuation.

P. 6, L. ~3-8: Again, the example presented here attributes control actions (increasing step width) to increasing mechanical impedance, which conflates again the contributions of passive mechanics and active control.

We have significantly edited the manuscript to remove references to the concept of mechanical impedance.

P. 6, L. 10 - "... uncertainty of external perturbations..." - This phrase again mixes (I think) at least 3 different concepts: "uncertainty", "perturbations" and (by reference to the prior paragraph) "stability". Each of these could potentially have multiple definitions depending on the system and/or context being studied - None of these have yet been clearly defined here.

We have removed the sentence referred to above as well as the term “uncertainty.” As mentioned earlier, we have made every attempt to remove vague and undefined terms.

We explicitly define “stability” as “the ability to reject or recover from small external perturbations and return to a steady state (13-17)” P3 L17-18

Since there are a variety of types of perturbations used in studies on gait stability, we have chosen to use this more generalized term and refer to a number of papers that review concepts and metrics of stability.

P. 6, L. ~28: Now the concepts of "energetics" and "local stability" are introduced - and again no clear definitions given. "Local stability" for example is a very specific type of "stability" (i.e., it is distinct from "global stability", "parametric stability", etc.). There are also different types of "local stability" with different operational definitions for different systems and contexts.

We have removed all references to the term “energetics” and instead refer only to “metabolic cost” as defined by the Cost of Transport (COT). We have described the specifics of our calculation of local stability in the Methods section P. 11 L17 - P12 L3. We refer to specific papers on measuring local stability during gait, where we hope readers can go for a more thorough discussion of a complex topic.

Remainder of Intro in General: The themes expressed in the comments above continue - I don't think it is helpful for me to single out each and every one. In general, the Intro needs work. The work presented here is clearly trying to tackle a larger problem (“balance” maybe?) that involves multiple interacting sub-components (“uncertainty”, “perturbations”, “stability”, “energetic cost”, etc.). For readers to be able to discern the contribution this work will make to the scientific literature, these terms and concepts need to be clearly defined (as much as possible) and a more clear and focused description of how they are inter-related is needed.

We have significantly edited the Introduction. The revision now emphasizes how people modulate foot placement in response to the Movement Amplification environment and the associated changes in metabolic cost. We believe the important outcomes of this work are presented in a much more clear and focused manner.

For example, the current Abstract and Intro include lots of discussion of “uncertainty” - But the “movement amplification” force field used in this study, in the end, does not even address this question because it is, by construction, “predictable”....

We have significantly edited the Introduction and removed all references to “uncertainty.”

INTRODUCTION / METHODS:

It is clear that the viscous force fields imposed here are “predictable” - as in, they *can* be computed / predicted, they are not random, etc. But it is not clear to me that just because they *can* be predicted (mathematically) that we human beings can indeed predict them - The long-standing work on viscous curl fields in reaching suggests that we can, at least in some contexts - But I think this deserves more careful consideration.

We agree that we have not in fact verified that participants formed an internal model of or predicted the Movement Amplification fields. However, we believe that this novel method of increasing the requirements to control lateral center of mass motion during walking presents new and significant information concerning the strategies that people use to control this motion. We have removed the speculation that people formed an internal model.

METHODS:

P. 13, L. ~3-17: It is not clear here what was done to confirm the findings. If anthropometrics “influence” the accelerations and “there were absolute differences” observed in those accelerations, then what if any analyses were done to determine if and/or the extent to which these observed “differences” could be attributed to anthropometrics (or other factors)?

We have added a reference to a previous manuscript in which we calculate anthropometrics, and, in particular, body mass does have an impact on the center of mass dynamics that occur in response to an applied external force using our cable robot during walking. We have not performed a similar analysis in

the current study. The current study seeks to understand the mechanisms, particularly changes in foot placement, that people use to control lateral COM motion during walking. The effects we observed were robust across all participants, thus minimizing the value of performing additional analyses on this topic.

P. 13, L. ~23-29: I don't understand this sentence. Just because participants were "asked" to control their COM does not mean they necessarily did as they were asked. This may just be a phrasing issue, but the experimental design cannot entirely "distinguish" what participants were / were not "able" to do from what they "chose" to do. Certainly, instructions and so forth can help reduce these potential confounds, but cannot get rid of them entirely.

We have slightly rephrased the sentence on P8 L13-15 to say "This task was designed to motivate participants to expend effort to control their COM, which helped us to distinguish whether changes in lateral COM motion were due to an *inability* to control COM motion as opposed to a deliberate choice to not control COM motion."

P. 14, L. ~8-10: It is mentioned here and else where that "participants had reached steady state". Several of these analyses depend on this assumption. In the study description (P. 12, L. ~14-19) it was "assumed" participants had reached steady state in the second trial. Was this verified / validated after-the-fact? If so, how?

This is our first publication on this particular type of force field, and we chose a long washout period to be conservative. 2 minutes is much longer than the time necessary to complete 14 steps, which was the mean time constant we observed for participants to return to steady-state after exposure to a damping field in previous experiments (Wu et al. 2017). In general, we did not observe any aftereffects of the Movement Amplification field, and participants did not report perceptions of aftereffects. We have included the following details on P7 L14-16 of the revision: "...the applied forces were removed and participants walked for an additional 2 minutes to wash out any residual effects of the force field, based on previous experiments showing aftereffects in a damping field (30)."

We also performed a linear regression analysis on the cost of transport data in the 2 minute "steady state period" with respect to time to see if there were any trends of increasing/decreasing metabolic cost. We found no significant non-zero slopes for any condition, which supports that subjects had reached steady state.

P. 14, L. 38 - "... data was...": This is a common mistake, but the word "data" is plural, not singular ("datum" is singular). This should be "... data were..." etc. [see e.g.,: <https://blog.apastyle.org/apastyle/2012/07/data-is-or-data-are.html>]. This should be fixed throughout the manuscript.

We have corrected all such usages where appropriate.

P. 14, L. ~44-47: Was there a specific algorithm used identify IC and TO events? Several have been published - If so, please cite. If some custom algorithm was derived, please describe and also justify why previously published methods were not able to be used here.

We have used the same algorithm as our previously published work, which we have cited in the revised Methods. We have also added slightly more detail to describe this method on P11 L2-3:

"Time of initial foot contact (IC) and toe-off (TO) events were identified for each step based on local extrema of fore-aft positions of the calcaneus and 2nd metatarsal markers."

P. 15, L. ~10-15: Calculation not clear. Integrating power to compute work is fine - normalizing by body weight is fine and typical - But normalizing by # of strides taken does not make sense. Net work done over what time interval? Assuming over each stride, then you would obtain net work *for that stride* - how many strides is irrelevant. If integrating over the entire trial (all 100 strides at once), then dividing by 100 might make sense - but this would give you only total net work done over the entire trial and no measures from which to assess stride-to-stride differences in net work...??.

We have edited the methods to clarify that we calculated the average net work per stride, not for a particular time length. We did not assess stride-to-stride differences in net work, but examined overall behavior assuming that participants had reached steady state during the Late Exposure period of the trial. All of the analysis code is provided in OSF for reference.

P. 15, L. 26: What does "Based on dynamical systems theory..." mean? Is there a competing theory for non-dynamical systems? What would a non "dynamical" system be anyway?: i.e., a system that exhibits no dynamics? Perhaps a rock? So would that mean the alternative to "dynamical systems theory" is a theory that all things are just rocks?

[Again - just trying to point out that terminology and careful definitions of terms does matter - This phrase "dynamical systems theory" is thrown around all the time - but almost never defined]

While we fully agree with the reviewer that referring to Dynamical systems theory here does not add anything to our manuscript, we would like to mention that this measure is derived from dynamical systems theory, which, in and of itself, clearly is a thing (see for instance http://www.scholarpedia.org/article/Dynamical_systems). We agree with the reviewer that we wouldn't know what "non dynamical systems" are, but then again, we are not direct experts in this field. Either way, we agree with the reviewer that the statement "based on dynamical systems theory" can be omitted in our manuscript, and have done so.

P. 15, L. ~24-45: Several published papers have also now shown that these algorithms for computing λ are also sensitive to a number of choices & parameters used in the analyses: definition of state space, choice of time delay, etc etc. How were the algorithmic choices made here validated?

According to the original work by Rosenstein¹, his algorithm is actually rather robust against using wrong embedding parameters. Nonetheless, the reviewer is right that later work has shown that parameter choices may significantly influence validity of these measures. We chose an embedding dimension of 5, and delay of 10 (after normalizing the time series such that an average stride had 100 samples, in line with our earlier work and^{2,3}), with the idea that, in this way, we had minimal influences of parameter choices (since these are similar between all subjects and conditions). These are often-used parameter choices⁴⁻⁹. In fact, van Schooten et al.¹⁰ showed that using fixed parameters for creating state spaces led to the smallest detectable difference, and highest within- and between- session reliability.

1. Rosenstein MT, Collins JJ, De Luca CJ. A practical method for calculating largest Lyapunov exponents from small data sets. *Physica D: Nonlinear Phenomena*. 1993;65(1-2):117-34.
2. Raffalt PC, Kent JA, Wurdeman SR, Stergiou N. Selection Procedures for the Largest Lyapunov Exponent in Gait Biomechanics. *Annals of biomedical engineering*. 2019 Apr 15;47(4):913-23.
3. Bruijn, S.M., et al., Assessing the stability of human locomotion: a review of current measures. *Journal of the Royal Society, Interface* 2013. 10(83): p. 20120999.
4. Bruijn SM, van Dieen JH, Meijer OG, Beek PJ. Statistical precision and sensitivity of measures of dynamic gait stability. *J Neurosci Methods*. 2009;178(2):327-33.
5. Bruijn SM, van Dieen JH, Meijer OG, Beek PJ. Is slow walking more stable? *J Biomech*. 2009;42(10):1506-12.

6. England SA, Granata KP. The influence of gait speed on local dynamic stability of walking. *Gait & posture*. 2007 Feb 1;25(2):172-8.
7. Buzzi UH, Stergiou N, Kurz MJ, Hageman PA, Heidel J. Nonlinear dynamics indicates aging affects variability during gait. *Clinical biomechanics*. 2003 Jun 1;18(5):435-43.
8. Dingwell JB, Cusumano JP. Nonlinear time series analysis of normal and pathological human walking. *Chaos: An Interdisciplinary Journal of Nonlinear Science*. 2000 Dec;10(4):848-63.
9. Dingwell JB, Cusumano JP, Cavanagh PR, Sternad D. Local dynamic stability versus kinematic variability of continuous overground and treadmill walking. *Journal of biomechanical engineering*. 2001 Feb 1;123(1):27-32.
10. van Schooten KS, Rispens SM, Pijnappels M, Daffertshofer A, van Dieen JH. Assessing gait stability: the influence of state space reconstruction on inter-and intra-day reliability of local dynamic stability during over-ground walking. *Journal of biomechanics*. 2013 Jan 4;46(1):137-41.

P. 15, L. ~46 - P. 16, L. ~20: As discussed above, the Intro discusses multiple concepts ("balance", "uncertainty", "perturbations", "stability", "energetic cost", etc.). These two paragraphs here now refer to "control" also - And again, this not clearly defined. First, the basic biomechanical measures (P. 15) are fine, but by themselves may or may not reflect "mechanisms used to implement COM control" - This is an inference or speculation that is being imposed by the authors on these dependent measures - But the measures themselves are not measures of "control" etc.

We agree that the kinematics-based metrics alone are not enough to make significant claims about control mechanisms used by the nervous system. We have made edits in the Methods section to remove claims about "control" mechanisms. We speculate on how these kinematic metrics relate to control strategies only in the Discussion of the revised manuscript.

We have edited the beginning of this section to emphasize that we are measuring kinematics to better understand gait strategies on P12 L4-5: "We calculated means and variability of several kinematic metrics to investigate the gait strategies employed in the different force fields."

P. 16, L. ~8-20: Likewise, this paragraph starts with the statement "to examine underlying control strategies...", but the metric (R^2 for regression of pelvis state to foot placement) is a correlation - And it is well known that correlation may or may not imply causation. So the notion of this as a measure of "control" is again an inference (i.e., speculation) by the authors. Indeed, their last sentence states that this R^2 metric can reflect "either increased feedback control or effects of passive dynamics" - So it is clearly not a measure of "control" per se.

We have removed the phrase "to examine underlying control strategies used" from the description of the calculation of the regression metric between COM dynamics and foot placement. We feel that the last sentence of this paragraph best explains that this metric describes the effects of either neural control or passive dynamics: "The R^2 value of the regression fit was a metric of the degree of coupling between foot placement and COM dynamics, with higher correlation suggesting either increased neural control or effects of passive dynamics (10)." (P12 L14-16)

P. 16, L. 37-40: The statement that "Net work could not be calculated..." is wrong. As the cables were "slack", the force applied by them was, by construction, zero - Therefore the net work was, by definition, also zero. Was this not why you did the first set of t-tests (L. ~33) to determine if Net Work was different from zero?

Yes, the reviewer is correct; we apologise for the confusion. We have edited the sentence in question to read, "Net Work was zero for the Null field since the Agility Trainer was turned off and the load cells attached in series hung slack." (P13 L1-2). The beginning of the paragraph also reminds the reader that

we are calculating "Net Work done by the field on the participant" (P12 L19). Yes, the reviewer is correct about the reason for performing the first set of t-tests in the statistical analysis.

Second - This statement is not related to "Statistical Analysis" so probably does not belong in this paragraph anyway.

We have chosen to include the statement in question under "Statistical Analysis" as to clarify why we conducted the t-tests against a value of zero.

P. 16., L. ~42-44: If Bonferroni corrections were used for 3 comparisons, then the significance level should not be $p < 0.05$, but $p < 0.05/3$, i.e., $p < 0.01667$.

We multiplied the p-values from all relevant statistical tests by the appropriate Bonferroni correction factor (e.g. 3) and then compared to a p-value of 0.05. This is equivalent to what the reviewer is describing above. We believe this is a matter of preference in how to present the results, and we have chosen to keep our method for simplicity (i.e. to avoid typing out all the corrected p-values for each set of comparisons).

P. 17, L. ~21-23: So Bonferroni corrections for the multiple comparisons between conditions for each measure is one issue... But there is a separate issue of Type I error related to the fact that these analyses were run on a large number of dependent measures (at least 9 acc. to L. ~10-15). Thus, the appropriate significance level should not be $p < 0.05$, but $p < 0.05/9$, i.e., $p < 0.00555$.

Yes, there are several dependent measures, but we were also testing several different hypotheses about stability, metabolic cost, and gait strategies. We do not believe it would have been appropriate to run a MANOVA with all metrics combined in order to consider Type I error, and we would have lost significant statistical power in doing so.

RESULTS:

"Participants": This does not seem to belong in "Results". The participants themselves are not the "results" - They are the participants from whom these results were obtained. Move to "Methods".

We have moved this section to the Methods.

Figure 4: What is not described in the text, and was not tested for statistically, but appear in the figure, are differences between Early and Late Exposure. In particular, it appears that from Early to Late, Net Work increased for the Low field, but decreased for the High field. A 2x2 ANOVA (i.e., Early/Late x Low/High) would be needed to confirm this. In particular, Fig. 4 looks like there might be a significant interaction effect here. If so, what would this mean? How would this be interpreted?

We did not perform a 2x2 ANOVA since we were not interested in interaction effects between exposure period (Early/Late) and field magnitude (Low/High). The interpretation of a possible interaction effect is indeed difficult and not directly relevant to our hypotheses. We were only interested in if the fields were significantly different from zero (i.e. providing positive or negative damping) and significantly different from each other (i.e. representing two different magnitude levels). Thus we performed two separate t-tests. We generally tried to choose statistical tests to be the minimum for the questions we were interested in.

Figure 5: So the title of this paper and the main narrative are that there is some "trade-off" between "stability" and metabolic cost (here, COT). But this figure appears to contradict that interpretation. There is a clear trend of increasing instability across conditions in Fig. 5a [& Stats confirmed that all of these differences were "significant"]. There is also a fairly clear trend increasing COT across conditions in Fig. 5b. No p-value is given for the Null vs. Low comparison, but even if this difference was not "statistically significant", this is exceptionally weak evidence of a "trade-off" b/t stability and COT. Indeed, the Discussion (P. 25, L. ~5-10) concedes as much: i.e., "*If* the two conditions were in fact equivalent, then participants *may* have chosen to sacrifice some stability... to maintain metabolic cost" [emphasis added] - This is a huge amount of hedging, especially when the authors' own equivalence test specifically could *not* establish the "equivalence" that the authors' primary conclusion is based upon.

We have significantly edited the revised manuscript, including the title, to de-emphasize trade-offs between stability and metabolic cost. Please see earlier responses to reviewer comments for more detailed discussions of this topic.

Instead, the overall result - looking at all of Fig. 5 together seems to far more clearly indicate the opposite of any such trade-off. Indeed, if data were pooled across conditions, and a direct correlation of λ_S to COT were performed, it seems highly likely this would yield a strong and *positive* (not negative) correlation.

We do not believe that it is valid to run a correlation analysis for only three data points (Null, Amplification Low, and Amplification High). Please see above responses to comments with regards to de-emphasizing the trade-off concept.

Figure 6 & the Corresponding Results: This figure is particularly difficult to 'decipher' as it mixes multiple variables that relate to multiple underlying constructs and also presents them in a somewhat random / haphazard order [e.g., why is step width variability (on panel (f)) not next to step width (in panel (b)), etc. - or, e.g., it might make sense to plot step widths (b) next to step lengths (d), but they are not - and/or, why are the more derived measures (MOS in (c), R^2 in (g)) thrown in with the basic stepping parameters they were derived from?].

We agree that the previous Figure 6 was confusing. We have separated out Figure 6 into two separate figures – one on mean values (Figure 5) and one on variability as well as R^2 correlation of COM dynamics and foot placement (Figure 6). We believe that the correlation of COM dynamics and foot placement should be grouped with variability metrics as it offers more insight on the structure of the variability (i.e. that the variability is correlated between two physically-related metrics and not just random variability).

Similarly, it is hard to see how R^2 (g) is conceptually related to step length variability (h) is related to mean DST's (i) - Yet, these 3 variables are plotted side-by-side as though the authors are expecting readers to draw such comparisons (even if only visually / qualitatively). In general, there just does not seem to be much rhyme-or-reason to this figure - It is not at all clear what overriding "result" this figure and/or these data are intended to portray. It is not at all clear what "question" these results are intended to answer.

Figures 5 and 6 are intended to answer questions about what types of control strategies might have been employed to result in changes in stability and metabolic cost. The revised Discussion interprets the results of these figures in the order of the subplots. In the revised Figure 5, we have plotted frontal plane kinematics in the first row to show the connection between COM speed, step width, and MOS

(which considers COM position and velocity as well as step width). The second row of figure 5 now contains mean step length and mean DST, which may be consistent with increased neural control. The plots in figure 6 further support either increased reliance on passive pendulum dynamics and/or increased neural control.

Overall, as currently presented, it seems the only real main "finding" is that when you expose people to these destabilizing lateral perturbations - they become more destabilized.

We believe that the original manuscript presented too many results in a disorganized manner that took away from the main hypotheses and conclusions. We believe that the revised version emphasizes the implications of our main finding that people modulate foot placement during walking to control their lateral COM motion and that controlling lateral COM motion incurs a metabolic energy cost.

DISCUSSION:

Sub-Section 1 - "Trade-offs between energetics and stability...": As "uncertainty" was not introduced nor manipulated in this experiment (there were no "certain" vs. "uncertain" conditions to compare), any discussions or conclusions about "uncertainty" seem out of place here, as they cannot be supported (either for or against) by any of the results presented.

We have significantly revised the Discussion. The concerns the reviewer raised have been removed.

Sub-Section 2 - "Differences in metabolic cost... may be explained by mechanical work": Again, this discussion, and any conclusions drawn from it are not really supported by the results, as currently presented. The authors have both Net Work data and metabolic cost (COT) data - So if how these are related is an important topic of this work, then direct correlations of these measures should be conducted to substantiate any claims regarding how they are related.

We believe that the correct correlation for this question would be between the mechanical work done *by the person* on their COM, which requires ground reaction force data to calculate, and metabolic cost. However, we were not able to collect GRF's in the current setup and mention this as future work in the last sentence of subsection 2.

Sub-Section 3 - "Biomechanical mechanisms suggest...": This section describes "control methods, including high impedance, increased feedback control, and/or decreased volitional control" etc. However, these constructs do not seem related to the actual data presented. none of the measures calculated are measures of "impedance" or of "feedback control" or of "volitional control" etc.

We have removed all references to impedance and feedback control in the revision and discuss only how our kinematic results are suggestive of various control methods. We have been careful to remove any statements that assert that these metrics directly point to any control methods.

Etc.

In General, the current Discussion covers many topics that stray quite far from the actual experiment that was implemented and the specific dependent measures computed and presented. As with the Introduction, what seems missing here is a clear (and preferably more linear) narrative (e.g., 'What is A? How does A lead to B? How then does B lead to C?', etc.).

We have significantly edited the Discussion to follow the narrative of the Introduction.

The Discussion and the manuscript as a whole could be very much improved if all of this ancillary and not-directly-relevant (and highly speculative) "stuff" was removed and the paper re-written to focus much more specifically on exactly what were the main, important, significant questions asked? - And exactly how does each result presented speak specifically to each of those questions? Etc.

We agree with the reviewer that the original manuscript required significant revisions and appreciate the detailed comments to refocus the paper. We have removed much of the not-directly-relevant and highly speculative statements from the revised Discussion and believe that this revised manuscript significantly shifts the focus to the main hypotheses that people in the Movement Amplification field will modulate their stepping patterns to aid in control of the lateral COM motion, and that the strategies to control lateral motion will increase the metabolic cost of transport.